# Seeking safety: Movement dynamics after post-contact immobility

**Nigel R. Franks[1]\*, Alan Worley[1], George T. Fortune[2], Raymond E. Goldstein[2], Ana B. Sendova-Franks[1]**

1 School of Biological Sciences, University of Bristol, Bristol, United Kingdom, 2 Department of Applied Mathematics and Theoretical Physics, Centre for Mathematical Sciences, University of Cambridge, Cambridge, United Kingdom

\* nigel.franks@bristol.ac.uk

**Data Availability Statement:** All relevant data are within the manuscript and its Supporting information files.

## Abstract

Post-contact immobility (PCI) is a final attempt to avoid predation. Here, for the first time, we examine the pattern of movement and immobility when antlion larvae resume activity after PCI. To simulate contact with, and escape from, a predator we dropped the larvae onto three different substrates: Paper, Shallow sand (2.3mm-deep) and Deep sand (4.6mm-deep). The Paper lining a Petri dish represented a hard surface that antlion larvae could not penetrate to hide. The Shallow sand permitted the antlions to dig but not to submerge completely whereas the Deep sand allowed them both to dig and to submerge. We tracked their paths automatically and recorded alternating immobility and movement durations over 90min. On the impenetrable substrate, antlion larvae showed super-diffusive dispersal, their movement durations became longer, their immobility durations became shorter and their instantaneous speeds increased. This is consistent with the antlions needing to leave an area of hard substrate and quickly to find somewhere to hide. On Shallow sand, antlion larvae exhibited a modest increase in movement duration, a modest decrease in immobility duration and a concomitant diffusive dispersal. This is consistent with their use of a spiral search, presumably for a suitable depth of sand, to conceal themselves. On Deep sand, the movement and immobility durations of the antlion larvae did not change and their dispersal was sub-diffusive because they were able to bury themselves. On Paper, the distribution of immobility durations had a long tail, consistent with a log-normal distribution. On Shallow and Deep sand, most of the distribution was fitted better by a power law or a log-normal. Our results suggest that PCI in antlion larvae is a disruptive event and that post-PCI movement and immobility gradually return to the pattern typical of intermittent locomotion, depending on the scope for burying and hiding in the substrate.

## Introduction

A wide diversity of animals, both vertebrates and invertebrates, become immobile for considerable periods after physical contact with a potential predator [1]. Such immobility has been recognized by a plethora of terms: thanatosis, playing possum, death-feigning and tonic

**Funding:** This work was supported in part by the Engineering and Physical Sciences Research Council through a doctoral training fellowship (G.T. F.) and the Schlumberger Chair Fund at DAMTP, University of Cambridge (R.E.G.). There was no additional external or internal funding received for this study. The funders had no role in study design, data collection and analysis, decision to publish, or preparation of the manuscript.

**Competing interests:** The authors have declared that no competing interests exist.

immobility, to name but a few [1–5]. Here for the first time, to our knowledge, we investigate what animals do after this inactive state.

Immobility after contact by a predator is a final attempt to avoid being eaten. It should be distinguished from another type of immobility, freezing [1], which occurs earlier in the interaction sequence between predator and prey, before any physical contact [6]. Heritable variation in immobility after predator contact has been demonstrated by artificial selection in flour beetles [7]. If selected for long immobility, these beetles had a four times lower predation rate when in the presence of a conspecific selected for short immobility than when alone [8]. We advocate the adoption of post-contact immobility (PCI) as an unambiguous, quantitative, and less loaded term to distinguish it from freezing and encourage a comparative methodology [9]. We have also suggested that progress in understanding such behaviour will be facilitated by studying the length of time large numbers of species-specific individuals remain immobile and the distributional properties of such immobility duration [9]. Both have been adopted by other investigators [10, 11].

Our model system involves larvae of the European antlion *Euroleon nostras* (Geoffroy in Fourcroy, 1785). They are known as ferocious sit-and-wait predators that build their pit traps in sandy soil [12–14]. However, antlion larvae themselves could be targeted by visually-oriented predators, such as ground feeding birds [15] or small mammals because of their very noticeable conical pits. Indeed, antlion larvae are among the items fed by hoopoe parents to their nestlings [16]. Antlion pits often occur in clusters, a setting likely to promote the evolution of PCI because time is of the essence. A predator is likely to incur an opportunity cost when tackling one prey item while others are in close proximity [1]. The periods antlion larvae spend in PCI, after having been accidentally dropped on the ground by a putative predator, can span more than three orders of magnitude, from a second to more than one hour [9]. Recently, PCI duration spanning more than three orders of magnitude was also shown for the most common antlion species in the Atlantic lowlands of Costa Rica [10]. This is consistent with earlier work on diverse species. Even though in many of these studies the distribution of PCI duration is not shown directly, its right skew is indicated by the Gaussian approximation after transformation with a square-root or a log [e.g. 2, 11]. Careful examination of the distribution of such immobility periods shows that it closely approximates an exponential [9], which has a long right tail and a standard deviation equal to the mean. This implies that the length of time that an antlion remains immobile is inherently unpredictable within a wide range. Thus, even though at the population level PCI has a totally predictable half-life (median value), a predator having dropped an antlion larva cannot estimate how long it will remain immobile and presumably undetectable. It might change state immediately or at any time in an endless future. The combination of a predictable average and an unpredictable individual PCI termination time makes PCI duration amenable to selection through predator-prey coevolution. Indeed, we recently used the marginal value theorem to demonstrate that PCI with a longer half-live facilitates survival when prey, such as antlions, occur in patches [17] and that antlions would not do substantially better if their half-lives were longer than we have established empirically [9, 17].

An increase in the mean of an exponential is accompanied by an increase in its variation and hence greater unpredictability in PCI duration. By contrast, distributions with a longer tail, such as a power law, do not have a characteristic scale and hence even the average PCI would be unpredictable. Indeed, the exponential distribution of PCI duration puts it at odds with the power-law distribution of immobility durations within intermittent locomotion [18] described in both vertebrates [19] and invertebrates [20]. An important question, therefore, is what happens to the pattern of movement and immobility when activity resumes after PCI. Animals that have experienced PCI must eventually go back to their usual

pattern of intermittent locomotion [18] but the characteristics of this transition might indicate a non-invasive method for recognising a less common behavioural and physiological state.

In an earlier study, we found that the first movement after PCI in antlion larvae was at most 1s-long and that the first immobility after PCI (the second immobility), although significantly shorter than PCI on average, was also closely approximated by an exponential distribution [9]. These results suggest that a power-law generative model, such as priority-based decision-making for choosing between movement and immobility [21, 22] would be wrong not only for the duration distribution of PCI itself but also for the movement and immobility immediately after the end of PCI. This is likely because the disruption caused by predator contact takes some time to attenuate and such disruption could be associated with a possible reduction in behavioural complexity [21, 23, 24]. However, as the time after the end of PCI increases and the associated disruption attenuates, the characteristics of spontaneous intermittent locomotion [18] might be restored. To explore this possibility, it is essential to examine the pattern of movement and immobility over time after the termination of PCI.

In the present study, we consider how potential prey might behave when their unpredictable period of post-contact immobility ends. Again, we use antlion larvae as our model system and induce PCI by dropping them onto the substrate as a simulation of contact by, and escape from, a putative predator [25] such as a ground-feeding bird or mammal [9, 17]. In addition to examining the sequence of movement and immobility durations after PCI, we also test whether they are context-dependent. More specifically, antlion larvae are burrowing sit-and-wait predators [12] and their preferences for substrate characteristics such as particle size [26, 27] and depth [28, 29] have been well documented. Indeed, recently it was demonstrated that antlion larvae display longer PCI duration when dropped on compact soil than on loose soil [10]. In other words, there is a trade-off between the protection conferred by PCI and the ease with which the antlion larva can hide itself in the substrate. Therefore, what the larvae do over time after PCI termination is likely to depend on the penetrability of the substrate. After emergence from PCI, submergence in the substrate is by far the best option for antlion larvae as the next step to safety. But what if this is not possible, such as when the putative predator drops the larva onto a stone surface?

We induced PCI on three different substrates: (1) Paper, lining a Petri dish, representing an impenetrable hard surface, such as the stone surface on which the predator might have inadvertently dropped the antlion; (2) Shallow sand, 2.3 mm, approximately the same depth as the 2 mm used in our earlier study [9], which permits some but not total submergence; (3) Deep sand, 4.6 mm, which allows complete submergence. Earlier studies on the effect of sand depth on pit building define 'shallow sand' and 'deep sand' as sand depths of 2 and 4 cm, respectively [29]. There is a large variation in the sand depths antlion larvae can experience [29]. Here we are using an order of magnitude shallower sand depths than these earlier studies because they allow only for partial or total body submergence but not for pit building. The objective is to simulate suboptimal conditions of the substrate on which the antlion might be dropped inadvertently by a putative predator. Such suboptimal conditions of different quality create a potential trade-off between seeking safety in PCI and seeking safety in substrate submergence. Following PCI induction, we recorded the first 16 alternating periods of immobility and movement, including PCI itself. This number was chosen based on the maximum number of immobility and movement periods we observed following PCI induction in antlions dropped on Paper before they reached the arena wall within the 90 min of observation. We examined the dynamics of movement and immobility after the induction of PCI in the context of the possibility or not of concealment in the substrate. Thus we could test whether antlions are seeking safety by behaving adaptively in a specific context.

Antlion larvae are likely to sense whether the substrate on which they have been dropped is hard or soft. On impact, they might do so through the rate of deceleration. When they begin to move after PCI, they might gauge hardness or softness through the resistance of the substrate. They are endowed with an evolutionarily fine-tuned ability to perceive vibrations by the hairs covering their bodies [30–32], which might also provide them with information about the substrate. Such reasoning is consistent with the recent demonstration that their PCI durations are longer on compact soil then on loose soil [10]. Hence, we hypothesise that antlions will respond differently to the hard surface of the Paper treatment compared to the two sand treatments. We predict that they search for the relative safety of escaping altogether from a hard substrate, since they cannot hide by digging. The question is how they balance the immediate danger from the putative predator that has dropped them against the future danger of being exposed on the substrate surface. So we ask what is their pattern of mobility and immobility after PCI? The difference in submergence possibilities in the two sand treatments is, however, less dramatic. Are antlion larvae able to perceive differences in sand depth and respond accordingly in terms of the dynamics of movement and immobility periods after PCI?

## Materials and methods

We received the approval of the University of Bristol Ethics of Research Committee for this study (University Investigator no. UN/19/006).

### Experimental subjects and procedure

Twenty-two *E. nostras* antlion larvae were collected from southwest Guernsey on 12 June 2019 and placed directly into individually-marked and empty vials. Each larva was weighed early the following morning (Smart Weigh GEM20, max weight 20g, precision 0.001g). Larval weight covered a wide range and likely represented the 2nd and 3rd larval stages (median = 0.0360 g, LQ = 0.0183 g, UQ = 0.0850 g, S1 Table). The experiments took place from 14 to 18 June 2019 (S1 Table) during daylight hours at a room temperature between 20 and 25°C [33]. When not taking part in experiments, antlions were kept in isolation in a dark and dry place in the same room. No food or water were provided at any time. The larvae are sit-and-wait predators that build their pits in places protected from rain and direct sunlight, spend most of their time in darkness at the bottom of their pits with only their mouthparts above the substrate, have a low metabolic rate [34] and can withstand long periods without sustenance [35]. After the experiment, all 22 antlion larvae were released on 19 June 2019 at the same site from which they were collected.

Experiments were carried out in two sessions (one in the morning and one at midday or in the early afternoon, to cover the room temperature range, S1 Table) with four antlion larvae at the same time, each in its own experimental arena: the bottom of a Petri dish with internal dimensions 225×225×18 mm (Corning® 431272). The four Petri dishes were arranged in a 2x2 grid and were filmed simultaneously from 75cm above with a Sony RX100 MK V camera on a Keiser copy stand by recording 1080P MP4 video at 50 fps. Filming continued for ~90 min on four consecutive clips of 22 min 31 s (the camera's time limit for continuous filming) with ~10 s in between. In each case, we compared the last frame of a clip to the first frame of the next clip and found no evidence of any displacement or change in body outline. An additional fifth clip of ~2 min duration included antlion identification labels placed on the arenas. Light was provided by four LED bulbs (dimmable, 9 W, 230 V, 806 Lumens, 2700 K colour temperature), one in each of the four corners of the frame.

The three experimental treatments corresponded to the three substrates on which the larvae were dropped: (i) Paper (a white sheet of A4 paper lining the Petri dish), (ii) Shallow sand,

2.3mm-deep sand (120 ml decorative white sand, Trustleaf Aquasand, Natural White Silica Sand, pH Neutral, grain diameter between 0.250 and 0.500 mm, poured onto the Petri dish; this is approximately the same as the 2mm-deep sand substrate in Sendova-Franks et al. [9]), (iii) Deep sand, 4.6mm-deep sand (240 ml of the same make of decorative white sand as for the Shallow-sand treatment, poured onto the Petri dish). A typical *E. nostras* 3rd instar larva is 4.4mm high (as measured by the distance between the highest point on the thorax and the lowest point on the coxa of its third leg, [36]). Hence, the Shallow sand allowed for partial submergence while the Deep sand allowed for complete submergence. We provided a fresh substrate surface for each antlion in each treatment. Thus a fresh sheet of paper and a fresh portion of 120ml sand were provided for each antlion on the Paper and Shallow-sand substrates. A fresh, top-up portion of 120ml white sand (out of the total 240ml) was provided for each antlion on the Deep-sand substrate. In all cases the poured sand was levelled by gently and repeatedly tilting the arena from side to side.

Before the start of each experimental session, the centre of the experimental arena (112.5 mm from each wall and 159 mm from each corner) was marked by a dot on the paper or by a small indentation in the sand. This was the spot where antlions were dropped from their vial from ~20 mm (~2 larva lengths) above the substrate. In each session, antlions were delivered from left to right on each row of the 2x2 grid, starting from the left-hand arena on the first row and finishing with right-hand arena on the second row.

The 22 antlion larvae were sorted by weight and divided in four strata of 5–6 individuals. One individual was randomly selected, without replacement, from each of these four strata to form the group of four allocated to each experimental session on the 2x2 grid. Such a stratified random sampling ensured each experimental session for each treatment was representative of the weight range. Sessions alternated between starting with either the heaviest or the lightest individual.

All 22 antlions were tested on Paper with the sixth session involving only two individuals, which were delivered to the left and right arena on the first row while the other two remained empty. Eight larvae were allocated to each of the two sand substrates in the same groups of four as for the Paper substrate but delivered in the reverse order (S1 Table). Therefore, the number of antlions for each substrate were: N = 22 (Paper), N = 8 (Shallow sand), N = 8 (Deep sand) and each antlion tested on one of the sand substrates (Shallow or Deep) was also tested on Paper. Such repeated design increases statistical power and thus reduces the chance of the small sample size leading to a Type II error, namely failure to reject the null hypothesis when it is false. Indeed, although *E. nostras* is the most common antlion species in Europe, it is rare in Guernsey and even more rare in the UK. The repeated design was reflected in statistical mixed models for analysis (see Statistical analyses). The six antlions that did not undergo a Shallow or Deep sand treatment took part in an experiment on a tilted sand substrate, which is outside the scope of the present study (Franks et al., unpublished).

The Shallow-sand and Deep-sand treatments were applied on 14th and 15th June 2019, respectively and the Paper treatment was applied on 16th to 18th June 2019 (S1 Table). Therefore, due to time constraints, the order of treatment was not randomised and not every antlion underwent each of the three treatments. However, PCI duration in this species is reproducible when measured in the same individuals within 1–3 days [9]. This means that over a few days, as in the present study, the distribution of PCI duration does not change. In addition, we did not find any evidence for a difference in the change of either immobility or movement duration over time on Paper between the eight antlions tested first on Deep sand and the eight antlions tested first on Shallow sand. Hence the effects of treatment and order of treatment were not confounded.

## Data collection

Data for antlions tested on Paper were collected from the video recording starting with the time an individual was dropped and ending with the time it reached the arena wall. Only antlions 10 and 13 stayed in the vicinity of the dropping location at the arena centre and did not reach the arena wall within the 90 min of filming. The data for these two individuals, also tested on Deep and Shallow sand, respectively, were included in all the analyses. Antlions that reached the arena wall in less than 90 min did not venture back into the arena centre; instead they moved a little along the wall or to the nearest corner and then stopped. For the 22 antlion larvae tested on Paper, the interval from the time they were dropped to the time they reached the arena wall (or, in the case of antlions 10 and 13, did not move further) featured a minimum of two and a maximum of 16 immobility and movement periods.

By contrast to their movement on Paper, none of the antlions tested on the two sand substrates moved as far as the arena wall within the 90 min of filming. Hence, to facilitate comparisons with the maximum of 16 immobility and movement periods for antlions tested on Paper we collected data from the video for, at most, the first 16 such periods for antlions tested on the two sand substrates. It was possible to collect data for 16 immobility and movement periods for all of the eight antlions tested on Shallow sand but only for five of the eight antlions tested on Deep sand (antlions 9, 14 and 17 had 12, 8 and 13 pairs of immobility and movement periods, respectively), which also illustrates the smaller amount of movement on Deep sand even compared to Shallow sand.

The start and end time of each movement period were collected manually with a precision of 1s during playback using standard video software in Windows 10. The start of a movement period was defined in the same way as the termination of PCI [9] and a movement period was considered to have terminated after no movement for more than a couple of seconds. From these start and end times, we calculated the duration of each of the, at most, 16 immobility and movement periods after dropping for each of the individuals tested on each of the three substrates (S1 Table). These manual measurements of movement and immobility duration were corroborated by the image analysis carried out later (see below, S6–S8 Figs).

In addition to extracting the duration of movement and immobility periods manually, we used bespoke image-analysis scripts for automated tracking of the antlion paths. On Paper, after cropping the images to separate the four arenas, we extracted the time (s) and the x- and y-coordinates (mm) of the centroid of each antlion at 10 fps. The first time point ranged between 37.75 and 91.75 s to ensure that the image analysis for each of the 22 antlions began after the delivering hand of the experimenter had been withdrawn from the camera's field of view for the whole session. These data were coarse-grained to avoid fictive records of movement at very short time intervals. We replaced every two successive values for each of the three variables by their average and repeated this recursively five times, each time halving the data. This process resulted in five levels of coarse-graining at time intervals of 0.2, 0.4, 0.8, 1.6 and 3.2 s, respectively. We checked that the timings of immobility and movement in the data at all five levels of coarse-graining matched the respective timings from the data collected manually and that spatial analysis gave similar results across different coarse-graining levels. The results presented are based on the 4x coarse-grained data, at 1.6s time intervals. Our exploratory analysis showed this level of coarse-graining to be a good compromise between retaining the features present at all coarse-graining levels and avoiding spurious movements at short intervals.

Automated image-analysis tracking of individual antlions on the two sand substrates was challenging because movement was associated with the antlion burying itself in the substrate. This included throwing sand, which masked the boundary between the antlion and the sand. On Deep sand, in particular, the length scale of antlion movement was comparable with the

length scale of the measurement error. We used additional procedures to extract the centre positions of the antlions using a combination of bespoke Matlab [37] scripts and the open source image processing program Fiji [38]. The presented results for the two sand substrates are based on coarse-graining at 1.6s intervals, as on Paper. Image analysis began after the delivering hand of the experimenter had been withdrawn from the camera's field of view, as on Paper. The first time point ranged between 35.2 and 81.6 s (4.8 and 92.8 s) for Shallow (Deep) sand. The sensitivity of tracking antlion movement on sand necessitated the removal of the first 1.6 to 6.4 s (0 to 4.8 s) at the beginning of each subsequent video for Shallow (Deep) sand to avoid any bias due to small camera jitter (associated with activating the camera). Furthermore, we smoothed the raw tracks on both sand substrates to minimise the stochastic noise caused by resolution issues. To achieve this smoothing, we calculated a running average.

The image-analysis tracking could detect the net movement of the antlion, in the coarse-grained footage, when such movement was greater than the resolution of the camera. This matched closely the movement recorded during direct observation of the video play-back.

Finally, we used the "Cell Counter" plugin [39] in ImageJ [40] to record the coordinates of the initial (at dropping) and final (at ~90 min) position of each antlion larva. On this basis we calculated the overall, start-to-finish, displacement for each individual on each of the three substrates.

## Statistical analyses

Statistical analyses, calculations and graphical representations were carried out with bespoke scripts written in R v. 4.3.2 [41]. We used the basic functions as well as the packages ggplot2 [42] and lattice [43] for graphics, lme4 [44] for Linear Mixed-effects Models (LMMs), lmerTest [45] and multcomp [46] for post-hoc tests, mgcv [47] for General Additive Mixed-effects Models (GAMMs), poweRlaw [48] for alternative fits to the power-law complementary cumulative distribution function (ccdf) and segmented [49] to compare the goodness-of-fit between broken-stick and simple linear regression models.

We fitted LMMs to test for any changes in immobility or movement duration with successive immobility or movement period after contact by the putative predator. The initial model for immobility or movement duration (s) included weight (g) as a covariate but it was not significant in either case, as with earlier results [9]. Adding the experimental session (Session 1: morning or Session 2: around midday, representing the daily temperature variation) as a random factor to the immobility or movement duration model explained zero variance and was not pursued further. We also tested models with different structure of the random predictor. The best model in each case was chosen using the AIC-minimisation criterion. The best fitting model for each of the log10-transformed immobility and movement duration (s) had as predictors log10(Sequential number), Treatment with three levels: Paper, Shallow sand and Deep sand, the interaction between the two and the random factor Antlion ID (to reflect the repeated experimental design). The only difference was that in the best model for immobility duration the random factor Antlion ID could vary around both the overall intercept and the slope while in the best model for movement duration it could vary only around the overall intercept. The random predictor Antlion ID was worth including in the modelling of each of immobility and movement duration because in each case the model including Antlion ID fitted the data better than the model without it (Log-likelihood test: $X^2_3 = 36.54$, P = $5.765*10^{-8}$ for immobility duration; $X^2_1 = 9.77$, P = 0.002 for movement duration). The LMMs fitted the data adequately (range and Shapiro-Wilks normality test for scaled residuals were -2.08 to 3.81, $W_{423} = 0.964$, P = $1.41*10^{-8}$ for immobility duration and -1.81 to 2.64, $W_{423} = 0.955$, P = $4.47*10^{-10}$ for movement duration). Although the small p-values mean the null hypothesis

that the residuals are consistent with a normal distribution is rejected, this is to be expected to some extent given that the sample sizes are very large (between two and 16 repeated measures for each of between eight and 22 antlions on each of three substrates). In addition, the LMMs were relatively easy to interpret because of the assumption that the relationship between immobility or movement duration and the successive number of immobility or movement period on a log-log scale is linear for each of the three substrates. Nevertheless, we checked whether the results of the LMMs were robust by fitting GAMMs for which the assumption of a linear relationship is relaxed. The p-values for the smooth (non-linear) terms in GAMMs are approximate and bootstrapped 95% confidence intervals for them were also calculated [47]. The GAMMs with the same structure as the LMMs fitted the data adequately ($R^2_{adj}$ = 19.2%, Deviance explained = 20.9%, n = 423 for immobility duration and $R^2_{adj}$ = 17.3%, Deviance explained = 18.4%, n = 423 for movement duration).

To test whether the distribution of immobility and movement durations followed an exponential or a longer-tailed distribution, such as the log-normal or the power law, we used the package for fitting alternatives to the power law based on the maximum-likelihood method and widely accepted techniques [50]. It involves a type of likelihood-ratio test, Vuong's test [50], to compare the goodness-of-fit of pairs of models with the null hypothesis that both models are equally far from the true distribution [48]. We fitted the continuous exponential, power law and log-normal cumulative probability distributions. To identify the best model, we also took into consideration the amount of discarded data in the estimation of the lower-bound value (xmin). In addition, we applied the parsimony principle. Hence, we chose the simpler model if both explained the data adequately. For example, the exponential distribution could be generated by a simpler model (a simple Poisson process) than the other two. Not all antlions displayed 16 immobility or 16 movement events within the experimental period (S1 Table) and hence some individuals were under-represented.

To measure the displacement of antlion larvae from the point where they were dropped to the position of their final movement within the observation period, we calculated the mean squared displacement (MSD) from the end of PCI to the end of the observation period. We used log-binned time by applying the data binning method of Christensen and Moloney [51, pp. 355–356] to mitigate the reduced sample sizes for later time points and to capture a gradual picture of any changes near the start. The overall observation duration of 90 min was divided into 14 intervals on a log scale and the mid-point for each interval was calculated as the geometric mean. Then we summed the mean squared displacements for all the antlions that moved within each log-time interval and divided this sum by their number. Finally, we plotted each value of the MSD and the upper limit of its 95% confidence interval (CI) against the geometric mean for the respective log-time interval on a log-log scale. The mean instantaneous speed (MIS) was calculated in an analogous way. We tested whether there was a sharp change in the dynamics of MSD and MIS by choosing between a segmented and a simple linear regression model for the change of MSD and MIS over time on each of the three substrates. The decision involved a test for the existence of one break point with an $H_o$ that there is no break point. The test P values for MSD on Paper, Shallow and Deep sand were 0.037, 0.005, $4.827*10^{-5}$, respectively and for MIS 0.893, 0.034, 0.197, respectively, with N = 10 in all cases (based on the 14 time intervals and the lost degrees of freedom due to the model fitting). On the basis of these results and for consistency, we fitted a segmented linear regression model to the MSD plots and a simple linear regression model to the MIS plots. The $R^2_{adj}$ for the MSD segmented linear regression model on Paper, Shallow and Deep sand was 95.1, 85.0, 79.1%, respectively and for the MIS simple linear regression model 30.6, 43.7, 70.5%, respectively. The assumption that the standardized residuals are normally distributed was met in all cases (Shapiro-Wilk normality test: $W_x$ = 0.95, P = 0.548, $W_x$ = 0.97, P = 0.915, $W_x$ = 0.95, P = 0.565 for

the MSD segmented linear regression model on Paper, Shallow and Deep sand, respectively, $W_x = 0.97$, $P = 0.872$, $W_x = 0.87$, $P = 0.047$, $W_x = 0.93$, $P = 0.325$ for the MIS simple linear regression model, respectively).

We fitted an LMM to the start-to-finish displacement (mm) to test for differences between substrates. The log10-transformed displacement was the response while Treatment with three levels: Paper, Shallow sand and Deep sand together with the random factor Antlion ID were the predictors. The scaled residuals ranged from -3.91 to 1.05, suggesting a left skew. Indeed, the distribution of scaled residuals was significantly different from normal (Shapiro-Wilks normality test: $W_{38} = 0.620$, $P = 1.07*10^{-8}$). This was mostly explained by the two outliers for antlions 10 and 13 that hardly moved on Paper, but this was not considered sufficient justification to remove these two individuals from the analysis on all substrates. The random factor predictor Antlion ID had to be included in the model, because antlions tested on Paper were also tested on Shallow or Deep sand, but it did not have a significant effect. The LMM did not fit the data better than its equivalent excluding Antlion ID ($X^2 = 0.03$, d.f. = 1, p = 0.856).

## Results

### Dynamics of immobility duration

On Paper and Shallow sand, the duration of antlion immobility decreased with each successive immobility event, but it did not change on Deep sand. The decrease on Paper and Shallow sand diminished over successive events. This relationship was described well by a linear model on a log-log scale (Fig 1). The slope was significantly different from zero for Paper (LMM, slope = -1.18, $t_{59.4} = -6.41$, $P = 2.60*10^{-8}$) and Shallow sand (LMM, slope = -1.00, $t_{46.5} = -4.58$, $P = 3.54*10^{-5}$) but not for Deep sand (LMM, slope = -0.36, $t_{51.8} = -1.59$, $P = 0.12$). It was significantly greater for Paper than for Deep sand but there was no significant difference between the slopes for Paper and Shallow sand and between Shallow and Deep sand (Table 1). The above relative differences between the three substrates were confirmed by a GAMM fit of splines on a log-log scale (S1 Fig). The P value IQR for the difference between Paper and Deep sand was ($6.0*10^{-4}$, 0.05) and for the difference between Shallow sand and Deep sand was ($9.7*10^{-3}$, 0.16).

### Dynamics of movement duration

On Paper and Shallow sand, the duration of antlion movement increased with each successive movement event but it did not change on Deep sand. The increase on Paper and Shallow sand escalated over successive events. This relationship was described well by a linear model on a log-log scale (Fig 2). The slope was significantly different from zero for Paper (LMM, slope = 1.16, $t_{417.0} = 9.37$, $P < 2*10^{-16}$) and Shallow (LMM, slope = 0.38, $t_{319.4} = 2.65$, $P = 8*10^{-3}$) sand but not for Deep sand (LMM, slope = 0.01, $t_{399.9} = 0.63$, $P = 0.95$). It was significantly greater for Paper than for either Shallow or Deep sand but there was no significant difference between the slopes for Shallow and Deep sand (Table 2). The above relative differences between the three substrates were confirmed by a GAMM fit of splines on a log-log scale (S2 Fig). The P value IQR for the difference between Shallow sand and Paper was ($7.7*10^{-5}$, $6.2*10^{-3}$) and for the difference between Deep sand and Paper was (0.0, $1.04*10^{-6}$).

### Distributions of immobility and movement duration

The distribution of immobility durations on Paper had a very long cut-off, the longest portion of which was described well by the log-normal distribution. Its fit, however, was not statistically significantly better than that of the exponential or the power-law distribution (Fig 3 and

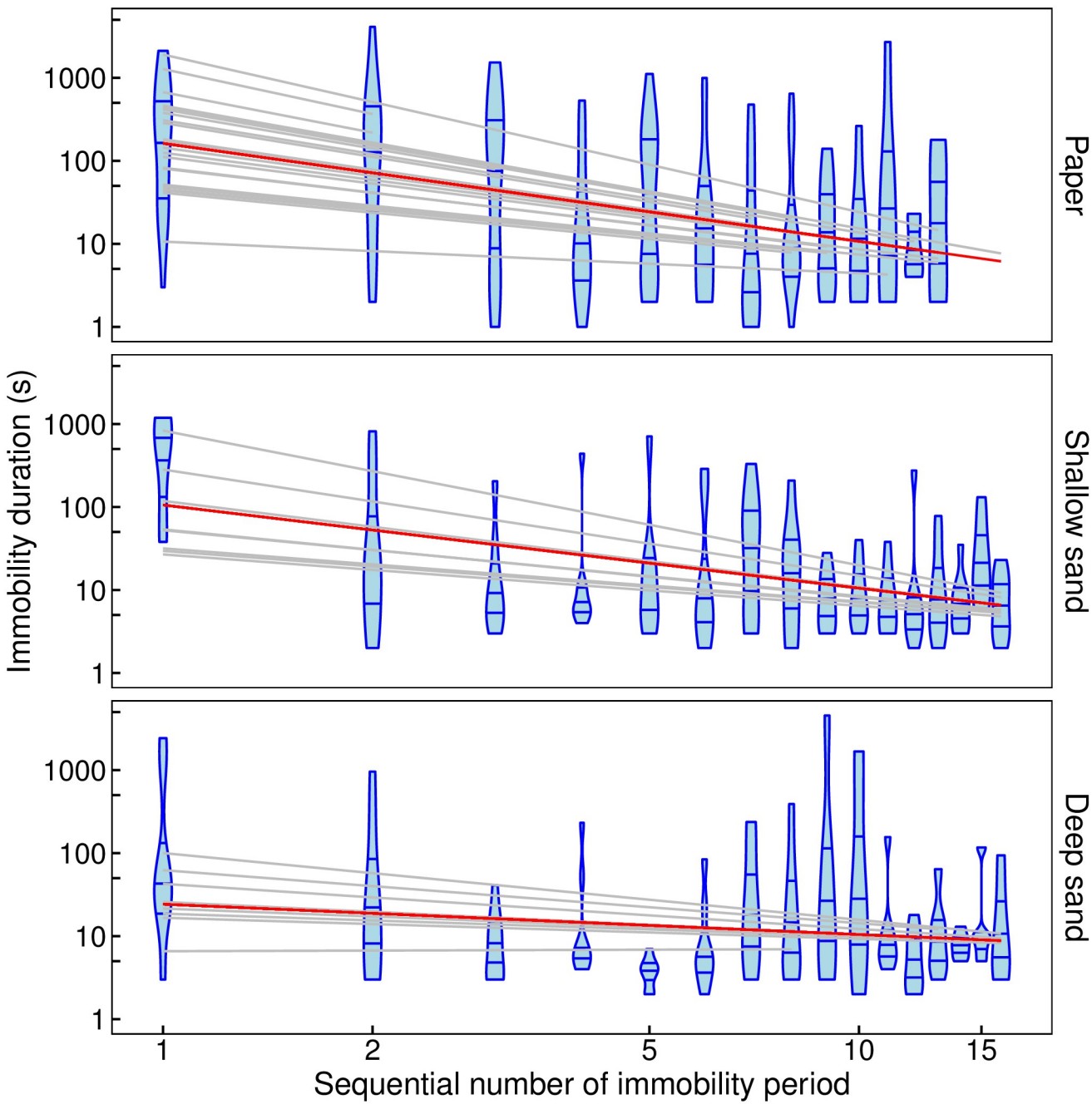

**Fig 1. Antlion immobility duration (s) against the sequential number of the immobility period since predator contact for each of the three substrates: Paper, Shallow sand, Deep sand.** Both axes are on a log scale; blue "violins": mirror density plots with horizontal lines representing the median, upper and lower quartile, red line: overall relationship (predicted fixed effects from the best LMM), grey lines: relationships for individual antlions (predicted random effects from the best LMM).

Table 3). By contrast, on Deep sand, where immobility durations did not decrease significantly with the increase in the number of events since contact with the putative predator as they did on Paper (Fig 1), the distribution of immobility durations was fitted well by the power-law and the log-normal over a long range. Furthermore, both of these fits were significantly better than

**Table 1. Post-hoc pair-wise comparisons between slopes for the best LMM for immobility duration (s).**

| Compared slopes | Estimated difference | SE | Z | P |
|---|---|---|---|---|
| Paper—Deep sand | 0.8159 | 0.2627 | 3.106 | 0.0053 |
| Shallow sand—Deep sand | 0.6376 | 0.2994 | 2.130 | 0.0832 |
| Paper—Shallow sand | 0.1783 | 0.2552 | 0.699 | 0.7631 |

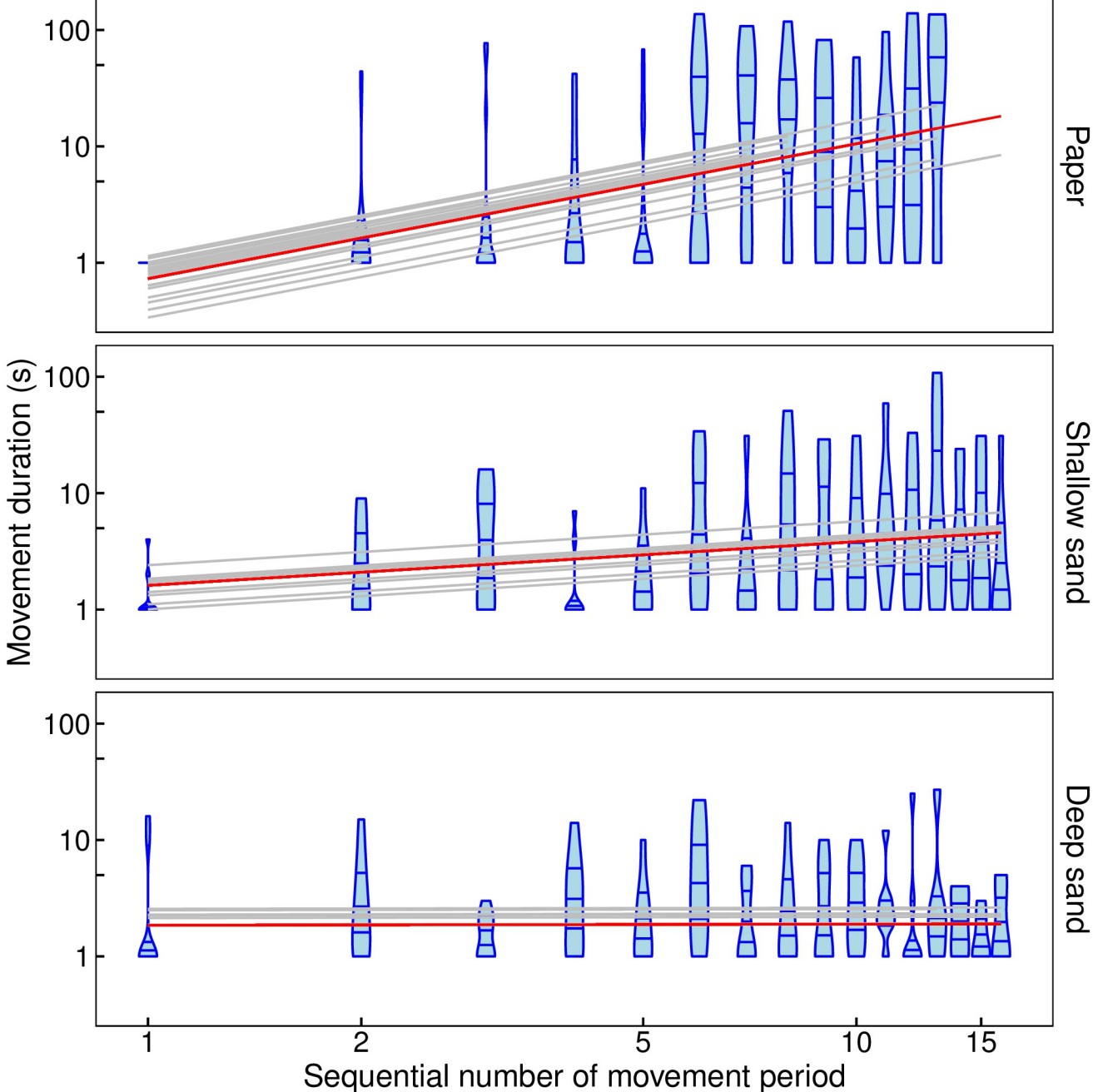

**Fig 2. Antlion movement duration (s) against the sequential number of the movement period since predator contact for each of the three substrates.**
Both axes are on a log scale; blue "violins": mirror density plots with horizontal lines representing the median, upper and lower quartile, red line: overall relationship (predicted fixed effects from the best LMM), grey lines: relationships for individual antlions (predicted random effects from the best LMM).

**Table 2. Post-hoc pair-wise comparisons between slopes for the best LMM for movement duration (s).**

| Compared slopes | Estimated difference | SE | Z | P |
|---|---|---|---|---|
| Shallow sand—Paper | -0.7836 | 0.1883 | -4.161 | 0.0001 |
| Deep sand—Paper | -1.1499 | 0.1962 | -5.862 | $<1 * 10^{-4}$ |
| Shallow sand—Deep sand | 0.3664 | 0.2077 | 1.764 | 0.1816 |

that of the exponential distribution, which fitted only the tail (Fig 3 and Table 3). The case of the Shallow sand, where there was a moderate decrease in immobility durations with each event (Fig 1), was intermediate: overall, both the power-law and the log-normal fitted the data significantly better than the exponential but the exponential fitted a longer tail than on Deep sand (Fig 3 and Table 3). Notably, the distributions for the two sand substrates overlapped considerably with the exception of the tails.

The relative pattern for the distribution of movement durations across the three substrates was similar to that for the distribution of immobility durations. However, they were all consistent with an exponential and there was no notable overlap between the distributions for the two sand substrates (Fig 4 and Table 4).

## Path, displacement and speed

As might be expected from the differences in movement duration between substrates, path length and the displacement of antlion larvae from the delivery spot in the centre of the experimental arena was substantial only on Paper (Fig 5a, 5c and 5e and S3–S5 Figs). The movement on both sand substrates involved different degrees of submergence but even the smallest larvae on Deep sand left an outline on the surface. Twenty of the 22 antlions tested on Paper reached the arena wall within the ~90 min of filming. Their median displacement from the arena centre, where they landed, to the arena wall (start-to-finish displacement) was 114.5 mm (the square arena's half-width and half-diagonal were 112.5 and 159 mm, respectively). By contrast, the horizontal displacement on the two sand substrates was minimal and none of the antlions reached the arena wall. Thus, on average antlions on Paper moved significantly further than antlions on either Shallow sand (median = 12.4 mm) or Deep sand (median = 4.1 mm, post-hoc test after LMM on log-transformed distance: P < 0.001 in both cases, Fig 6). Although, the difference in start-to-finish displacement between the two sand substrates was not significant (post-hoc test after LMM on log-transformed distance: P = 0.088), the interquartile ranges of the two distributions did not overlap (Fig 6).

The significantly greater displacement on paper was confirmed by the dispersion rate of antlion larvae on the three substrates. The mean squared displacement (MSD) on Paper was consistent with diffusion or even super-diffusion (95% CI for slope fitting most points straddles 1 but is mostly above it, Fig 7a), with diffusion on Shallow sand (95% CI for first slope straddles 1, Fig 7c) and with sub-diffusion on Deep sand (95% CI for first slope is below 1, Fig 7e). The relative order of the three substrates according to the rate at which antlion larvae dispersed on them was also reflected in the antlion speeds. On Paper, the mean instantaneous speed (MIS) increased with an approximately constant positive acceleration (95% CI for slope is above 0, Fig 7b, see also Fig 5b and S6 Fig). By contrast, on Shallow and, even more strongly, on Deep sand, the MIS decreased with an approximately constant deceleration (95% CI for slope is below 0, Fig 7d and 7f, see also Fig 5d and 5f and S7 and S8 Figs).

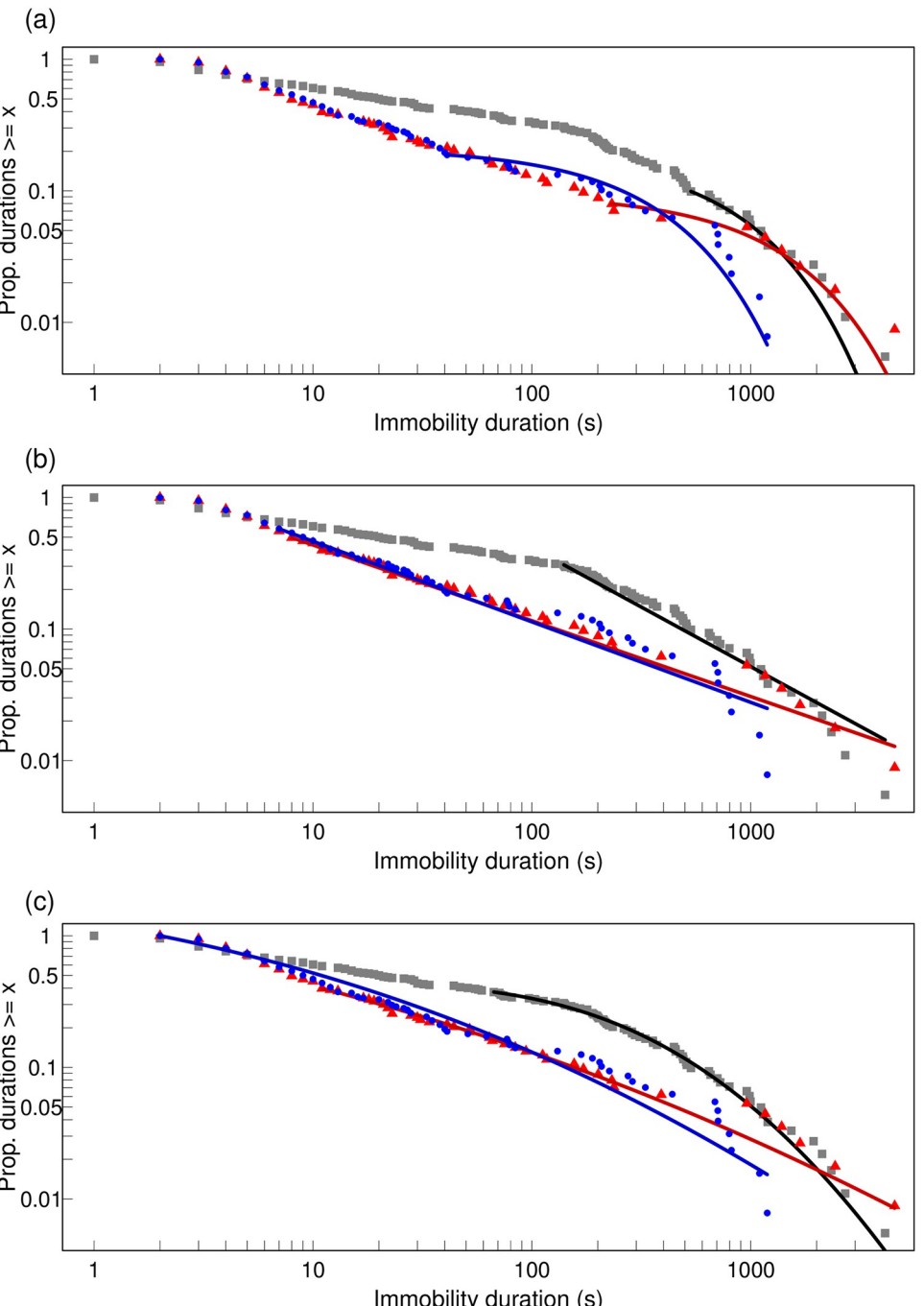

**Fig 3.** Empirical complementary cumulative distribution for immobility duration (s) for each of the three substrates fitted with each of the models: (a) Exponential, (b) Power law (c) Log-normal; grey square: data for Paper, blue circle: data for Shallow sand, red triangle: data for Deep sand; black line: fit for Paper; blue line: fit for Shallow sand; red line: fit for Deep sand. Both axes are on a log scale. Here each model is fitted with its own best-fit lower-bound (xmin) value to show how much of the data were discarded in each case but for the pair-wise comparisons with Vuong's test, xmin was the same for the members of each pair of compared models.

**Table 3. Complementary cumulative distribution for immobility duration (s): Goodness-of-fit tests comparing the exponential with alternatives.**

| Substrate, sample size | Vuong's test: EXP vs LN Statistic, P | Vuong's test: EXP vs PL Statistic, P | Vuong's test: LN vs PL Statistic, P |
|---|---|---|---|
| Paper, N = 182 | -1.737, 0.082 | -0.748, 0.455 | 1.193, 0.233 |
| Shallow sand, N = 128 | **-8.261, <0.001** | **-5.067, <0.001** | -0.105, 0.916 |
| Deep sand, N = 113 | **-4.222, <0.001** | **-4.754, <0.001** | -0.078, 0.937 |

A positive value for Vuong's test indicates first model is closer to true model; bold: second model is significantly closer to true model after a Bonferroni correction for multiple testing with α' = α/n = 0.05/3 = 0.017; p-value is two-sided; EXP: continuous exponential; LN: continuous log-normal; PW: continuous power law; the lower bound (xmin) for the comparison between EXP and LN is that for LN while for the comparison between EXP and PL as well as between LN and PL is for PL.

## Discussion

We found that, when antlion larvae terminate post-contact-immobility (PCI), they modify their pattern of movement and immobility to the opportunities for hiding afforded by the substrate on which they have been dropped by a putative predator. On Paper, with no possibility for submergence, immobility duration decreased and movement duration increased as a power law with successive immobility or movement events, respectively. On the Shallow sand, with the potential for only partial submergence, there was a similar but slower power-law decrease and an increase in immobility and movement duration, respectively. By contrast, given the potential for full submergence in the Deep sand, antlions that landed on this substrate did not show any change in the duration of either immobility or movement with successive immobility or movement events, respectively, after predator contact. This pattern of differences in response to different substrates was also expressed in the distribution of immobility duration. When the antlion larva landed on Paper, it had a long tail which was not significantly better fitted by the log-normal or the power law than the exponential. By contrast, when the antlion landed on sand, and on Deep sand in particular, the exponential fitted the distribution of immobility duration in a much shorter tail, while the power law and the log-normal fitted the data significantly better over the rest of the range.

When the antlion larvae could not escape, by burrowing, on Paper because the substrate was hard, their behaviour adapted to facilitate a horizontal escape instead. Their dispersion was super-diffusive and their instantaneous speed increased with time. The spatial characteristics of antlion movement after PCI on Shallow and Deep sand corresponded to their ability to hide in these two substrates. On Shallow sand, their dispersion was diffusive and their instantaneous speed decreased over time. This is consistent with the use of spiral search presumably for a suitable depth of sand where the larvae can conceal themselves completely because Shallow sand allowed only for partial submergence. By contrast, on Deep sand, where most of the antlions managed to submerge themselves completely, their dispersion was sub-diffusive and their instantaneous speed decreased over time more sharply than on Shallow sand. In complete concordance with the rate of their dispersion from the spot where they were dropped, the start-to-finish distance covered by the antlions was significantly longer on Paper than on either depth of sand. Furthermore, our results suggest it was also longer on Shallow sand than on Deep sand.

Due to time limitations and the small size of the sampled population, each of the two sand treatments involved eight antlions. Low sample sizes lead to low statistical power and a higher

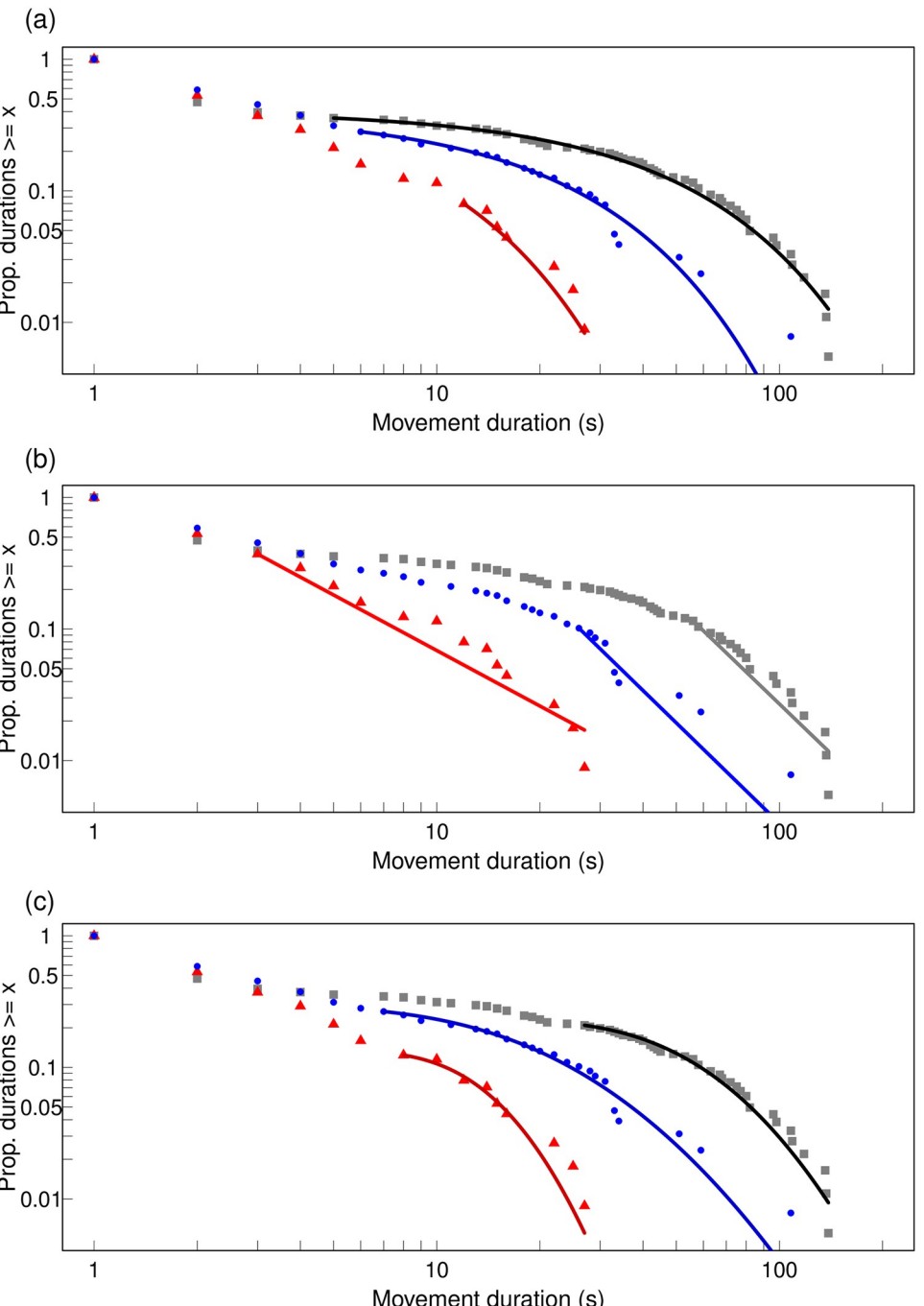

**Fig 4.** Empirical complementary cumulative distribution for movement duration (s) for each of the three substrates fitted with each of the models: (a) Exponential, (b) Power law (c) Log-normal; grey square: data for Paper, blue circle: data for Shallow sand, red triangle: data for Deep sand; black line: fit for Paper; blue line: fit for Shallow sand; red line: fit for Deep sand. Both axes are on a log scale. Here each model is fitted with its own best-fit lower-bound (xmin) value to show how much of the data were discarded in each case but for the pair-wise comparisons with Vuong's test, xmin was the same for the members of each pair of compared models.

**Table 4. Complementary cumulative distribution for movement duration (s): Goodness-of-fit tests comparing the exponential with alternatives.**

| Substrate, sample size | Vuong's test: EXP vs LN Statistic, P | Vuong's test: EXP vs PL Statistic, P | Vuong's test: LN vs PL Statistic, P |
|---|---|---|---|
| Paper, N = 182 | 0.181, 0.856 | 0.896, 0.370 | 0.238, 0.812 |
| Shallow sand, N = 128 | 0.655, 0.512 | -0.821, 0.411 | -0.214, 0.830 |
| Deep sand, N = 113 | 0.107, 0.915 | -1.087, 0.277 | -1.144, 0.253 |

A positive value for Vuong's test indicates first model is closer to true model; none of the models is significantly closer to true model before or after a Bonferroni correction for multiple testing with $\alpha' = \alpha/n = 0.05/3 = 0.017$; p-value is two-sided; EXP: continuous exponential; LN: continuous log-normal; PW: continuous power law; the lower bound (xmin) for the comparison between EXP and LN is that for LN while for the comparison between EXP and PL as well as between LN and PL is for PL.

risk of Type II error or a false negative result. Therefore, despite the multiple data points over time and space for each of these eight individuals and the paired design with the treatment on a hard surface, we are likely to have demonstrated more with a bigger sample size. However, although antlions are rare on the British Isles where we studied them, the same species, *Euroleon nostras* is abundant in continental Europe. We hope our results will encourage further studies in the same area with bigger sample sizes and more statistical power. Indeed, our study is exploratory and the first to investigate what happens after PCI. This is not a field where hundreds of sequential hypotheses, the majority of which are false, are tested and as such the low sample sizes for the two sand substrates are unlikely to incur a Type I error or a false positive result [52].

Again due to time limitations, we did not standardize the feeding state of the studied antlions even though the level of starvation could affect the behaviour or arthropods [53]. However, a disproportionate representation of feeding state is unlikely in the two sand treatments due to the smaller sample size because we selected the eight antlions for each as a stratified random sample after sorting all the 22 collected individuals according to weight. This was corroborated by a comparison between the behaviour of the eight antlions tested on Shallow sand and the eight tested on Deep sand when all 16 were tested on Paper. We found no differences between them in the change of immobility or movement duration with successive immobility or movement period (corresponding to Figs 1 and 2, respectively).

Our finding that there is a power-law relationship between both immobility and movement duration and the successive number of the event, particularly on Paper and Shallow Sand, suggests an underlying continuous feedback response by the antlions to these two substrates. Indeed, the power law is one of the mathematical functions used to describe learning curves in diverse species [54–56]. However, PCI durations are reproducible when measured again after a day of two in the same individuals [9], and if learning is involved in the feedback mechanism, it must apply to the durations of immobility and movement after the induction of PCI, not to PCI duration after successive inductions.

The above relationship between immobility and movement duration and the successive number of the event together with our results on the overall distribution of immobility duration represent quantitative measurements of the pattern of behavioural changes likely caused by predator contact. Henceforth, we will refer to this syndrome of changes as a disruption response [21]. Our results are objective evidence not only that such disruption exists but also that it is context-dependent: its magnitude was dependent on the extent to which the danger of predation [15, 16] was mitigated by a better opportunity to hide.

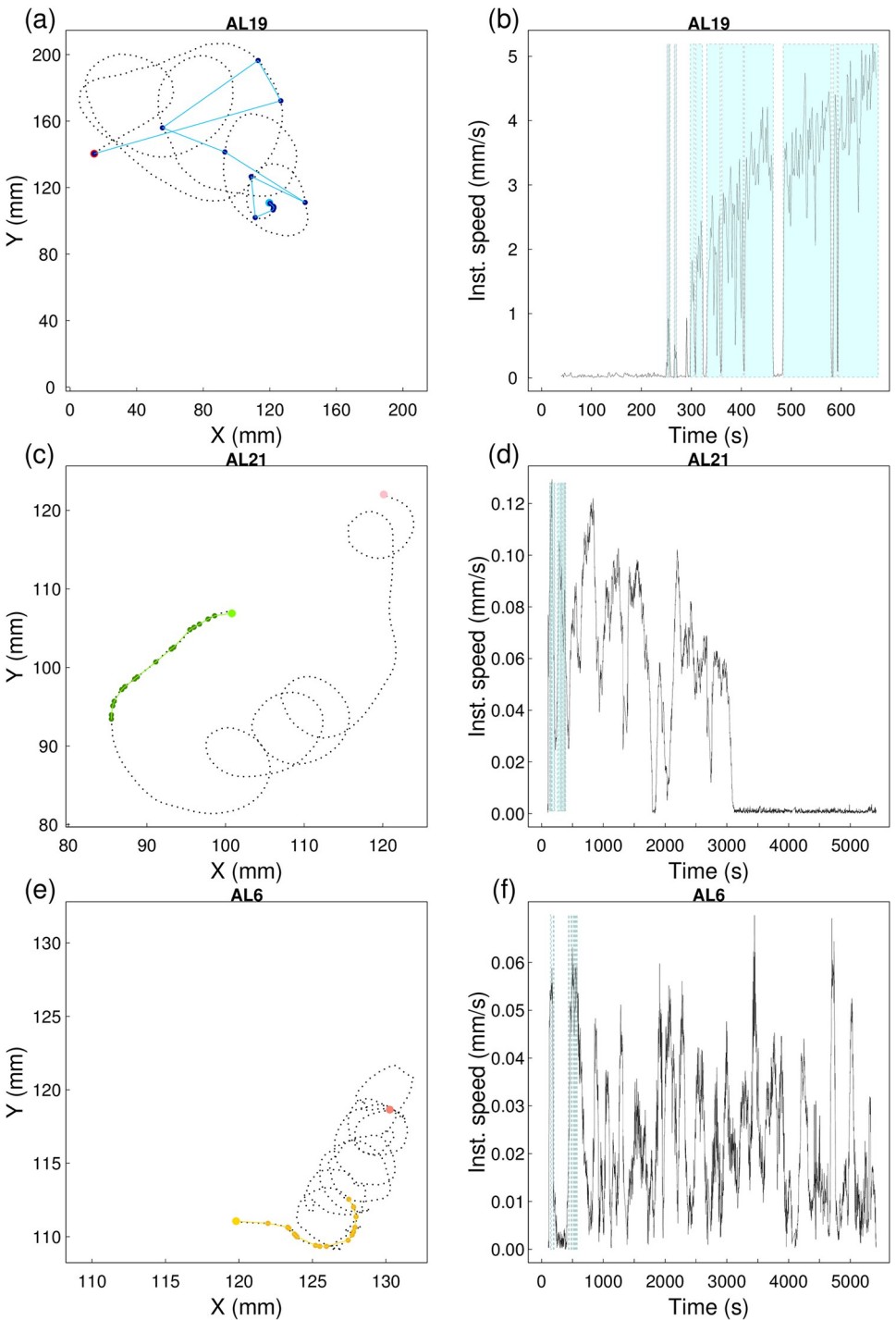

**Fig 5. Path (mm) and instantaneous speed (mms⁻¹) over time for one of the antlions on each of the three substrates.** (a)-(b) Antlion 19 on Paper, (c)-(d) Antlion 21 on Shallow sand and (e)-(f) Antlion 6 on Deep sand; dotted black line: path from the spot in the centre of the arena where the antlion larva was dropped (Paper: light blue circle; Shallow sand: light green circle; Deep sand: light yellow circle) to the wall (Paper: red circle) or end of up to 16 moves (Shallow sand: pink circle; Deep sand: salmon pink circle) with superimposed stops (Paper: dark blue circle; Shallow sand: dark green circle; Deep sand: dark yellow circle) and displacement segments between stops (Paper: light blue line; Shallow sand: light green line; Deep sand: light yellow line); black line: instantaneous speed over time (s) calculated from the image-analysis track, cyan highlight: the superimposed observed movement periods up to reaching the wall or end of up to 16 moves.

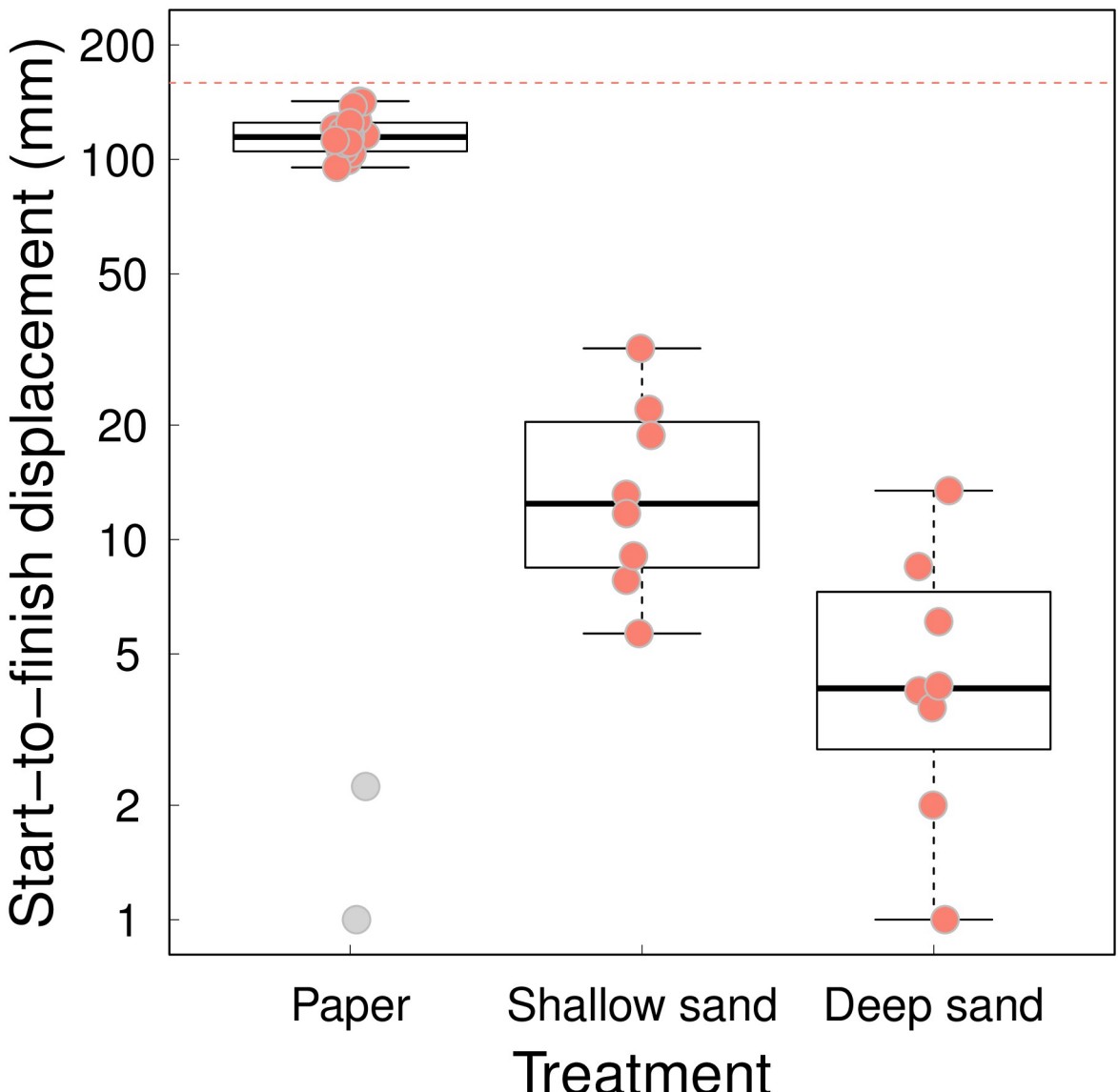

**Fig 6. Start-to-finish displacement (mm) for antlions on each of the three substrates: Paper (N = 22), Shallow sand (N = 8), Deep sand (N = 8).** The horizontal interrupted red line at 159 mm is half the diagonal of the experimental arena and represents the maximum possible start-to-finish displacement given that the antlions were delivered in the arena centre. Box plots show 25% and 75% quartiles (boxes), medians (lines in the boxes), outermost values within the range of 1.5 times the respective quartiles (whiskers), outliers (grey circles), and individual measurements (pink circles). A small amount of jitter was applied to the circles to minimise any occlusion. The two outliers represent antlions 10 and 13, that hardly moved throughout the experiment on Paper.

Earlier studies have demonstrated that anti-predator defensive behaviour is context-dependent in the sense that prey species are able to assess and adjust their response to the magnitude of the predation threat [57]. For example, PCI is exhibited more frequently and for longer durations when the level of predation threat is perceived to be higher by the larvae of the damselfly *Ischnura elegans* [57] and by the trashline orb-weaving spiders, *Cyclosa turbinate*, which is more at risk from visually-oriented predators during the day [58]. Another example of context-dependence is the shorter PCI duration when it occurs together with autohaemorrhaging

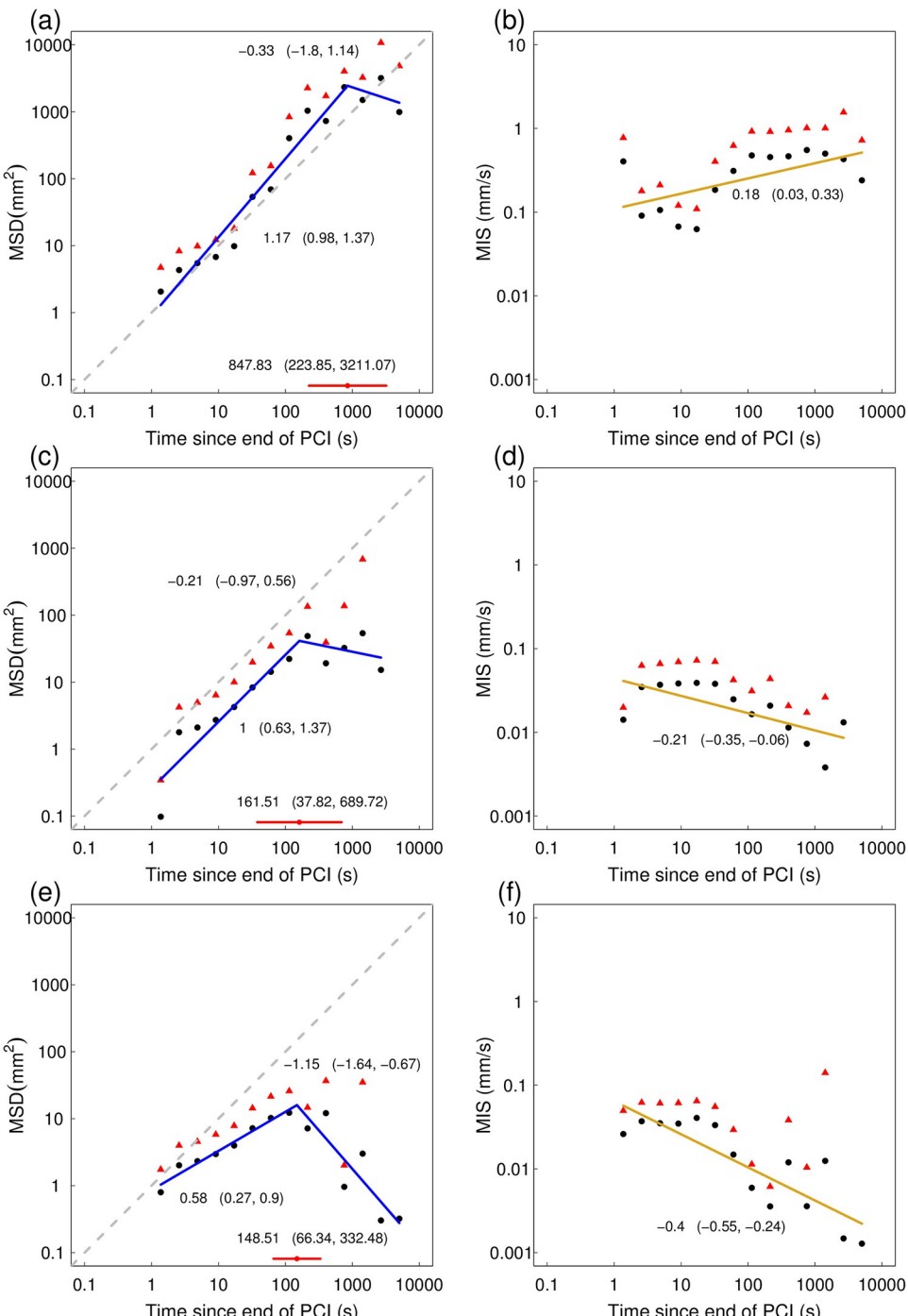

**Fig 7. Dispersion rate of antlion larvae on the three substrates Paper (N = 22), Shallow sand (N = 8), Deep sand (N = 8).** (a)-(b) mean squared displacement (MSD, mm$^2$) and mean instantaneous speed (MIS, mms$^{-1}$) over time since the end of PCI on Paper; (c)-(d) MSD and MIS on Shallow sand; (e)-(f) MSD and MIS on Deep sand; filled black circles: mean value for each of the 14 log-binned time intervals (see Methods for calculation details), red triangle: upper 95% CI limit of the mean for the time interval, solid blue line: line of best-fit from a segmented linear regression model with one break point, with estimates for the two slopes and their 95% CIs in brackets, red point and red horizontal line: estimate and 95% CI for the break point, interrupted grey line: the line of unity (y = x), solid yellow line: line of best fit from a simple linear regression with estimate and 95% CI for the slope; the number of mean values for Shallow sand is 13 and the number of upper 95% CI limits of the mean for both Shallow and Deep sand is 12 rather than 14, because there were fewer than two entries for the last two intervals for both sand substrates (see Methods for details of time binning).

in dice snakes [59]. Our novel contribution is to demonstrate that not only PCI itself, but what happens after PCI is context-dependent.

The antlion larvae showed a long-lasting response when they landed on a substrate that was too hard for them to burrow and hide, undetectable when they landed on sufficiently deep sand to disappear and intermediate when the sand allowed them to dig but not to hide fully. In addition, the overall distribution of immobility duration was most dissimilar to "normal" on Paper and most similar on Deep sand. Indeed, the intermittent locomotion of diverse organisms is characterised by power-law distributions of their immobility periods [19, 20, 60, 61] and, in many cases, of the movement periods too [20], even in the absence of environmental stimuli [62, 63]. Furthermore, such "waiting times" between activities [22] follow a power-law distribution not only at the behavioural level but also at other levels of biological organisation, such as genetic regulatory networks, morphogenesis, neural activity [64], human heart-rate [65, 66] or gate dynamics [67]. According to the theory of self-organised criticality [68], the lack of a characteristic scale, symptomatic of a power-law distribution, is a sign of the robustness and flexibility necessary for adaptive responses to new conditions [64]. Conversely, a breakdown in these scale-free characteristics is a symptom of illness, injury or distress, such as depression [69], heart malfunction or a stroke [23, 24] in humans. In a recent paper [9], we hypothesised that the unusual exponential distribution of PCI is a sign of distress, or as we call it more cautiously here, disruption. Such a reduction in complexity [21] is a response to the predator contact, but it may also represent a deficit in flexibility to other immediate changes. Our present results lend support to this hypothesis.

The quantitative measurement of stress at the behavioural level is a major aim in animal welfare [70]. Behavioural correlates of physiological responses are crucial for identifying stressful situations quickly. Methods such as the reduction in the fractal dimension of a behavioural frequency have been used to measure the effect of stressful events such as pregnancy and parasitic infection in Spanish ibex [71], and nematode parasitism in Japanese macaques [72]. To the best of our knowledge, the present study is the first to provide evidence of behavioural disruption, akin to such distress, at the behavioural level through changes in the distribution of immobility durations and in the context of predator-prey interactions.

The best means for an antlion larva to hide when inadvertently dropped by a putative predator is to bury itself in the sand. Intriguingly, when the antlions in our experiments were faced with one of two depths of sand or the solid surface of a Petri dish lined with paper, their response was not simply binary. In other words, they did not simply exhibit one pattern of intermittent locomotion when the substrate allowed partial or complete submergence and another pattern when no submergence was possible. Instead, our results suggest that antlion larvae may perceive environmental stimuli continuously and discriminate not simply between sand and a hard substrate, and hence between the presence and absence of an opportunity for submergence, but also between different depths of sand, and hence between the potential for different levels of submergence. The simplest candidate for a stimulus that would allow for such discrimination is the amount of resistance antlions encounter during locomotion, with least resistance on Paper and most resistance on Deep sand. This is plausible because, as sit-and-wait predators, antlion larvae have a multitude of hairs covering their bodies that provide them with sensory feedback [30–32]. They are literally bristling with information. Indeed, a recent study demonstrated that antlion larvae have a different PCI duration on different substrates [10]. This also appears to be suggested by our data but the differences were not statistically significant, possibly due to a smaller sample size, and were not reported. However, put together the results from both studies suggest that antlion larvae are able to perceive environmental stimuli from the very moment they land onto the substrate.

The highly quantitative approach we employ in the present study should provide a framework for understanding the strategy and physiological underpinnings of PCI in diverse animals. It has the potential to link our understanding of prey and predators, of vertebrates and invertebrates. We hope this framework will encourage others to study PCI duration in diverse organisms and to investigate what their model organisms do after emergence from PCI. Indeed, in some species PCI does not always occur after a stimulus and its frequency is likely to play a role in anti-predator tactics. For example, in the sexually dimorphic beetle *Gnathucerus cornutus*, the males have weaponry and PCI is more frequent in males with larger weapons [73]. By contrast, in the sexually dimorphic neotropical harvestman *Mischonyx cuspidatus*, PCI is less frequent in the male sex, where weaponry is present than in the female sex, where weaponry is absent [74]. Such findings call for comparisons of the movement and immobility durations when PCI occurs and when it does not occur. Behavioural studies of this type could be ideal for understanding arms races between predators and prey. In that sense, we think that there is a tremendous need also to study the behaviour of predators in the context both of PCI and what happens after PCI. This might be done through video-gaming scenarios so that prey are not harmed.

More generally, behavioural strategies involving a special bout of immobility extend beyond PCI. For example, they play a role in 'freezing', during the earlier stages of predator-prey interactions with the same predator species [1] or as the main strategy against a specific predator species [75]. Furthermore, they are sometimes manifest in sexual encounters. For example, in spiders, in which males provide nuptial gifts, males that also feign death gain more and longer copulations [76] while in dragonflies, females that 'drop and stop' effectively avoid the attention of additional males [77]. Last but not least, there could be a trade-off between the time invested in sexual behaviour and PCI duration as in the males of some weevil beetles, which spend less time in PCI when they would otherwise be searching for a mate [78]. In all such cases, a quantitative approach examining the immobility phenomenon within the dynamics of movement and immobility during intermittent locomotion [9] and the possibilities for escape afforded by the environment, as exemplified by the present study, should lead to a better understanding of both the tactics and strategies underpinning such behaviours.

## Supporting information

**S1 Table. Data on movement and immobility duration (s) for antlions on the three substrates: Paper, Shallow sand and Deep sand (uploaded as a separate excel file in the SI due to its size).** ALid: antlion identification number, Weight_g: antlion weight (g), Date: date of experiment; Sess: session 1 (in the morning) or session 2 (at midday or in the early afternoon) of the experiment on a given date, Treat: treatment substrate—Paper, Shallow sand (2.3sand) or Deep sand (4.6sand), Quad: quadrant in the 2x2 grid of four Petri dishes filmed simultaneously in which the antlion was dropped, Videos: number of videos of 22min31s duration with 10s-intervals between them that covered the first up to 16 immobility and movement periods for the antlion or up until it reached the arena wall, ArrT_s: arrival time (s) or the time when the antlion landed onto the substrate after being dropped from its individual vial, StTM1_s: start time for movement 1 (and similarly for all subsequent movement periods up to movement 16), I1dur_s: duration (s) of immobility period 1 (and similarly for all subsequent immobility periods up to immobility 16), EnTM1_s: end time for movement 1 (and similarly for all subsequent movement periods up to movement 16), M1dur_s: duration (s) of movement period 1 (and similarly for all subsequent movement periods up to movement 16). For further information, please see the Data Dictionary file.
(XLSX)

**S1 Fig. Antlion immobility duration (s) against the sequential number of the immobility period since predator contact for each of the three substrates: Paper, Shallow sand, Deep sand.** It is the same as Fig 1 except that the fitted lines are not based on the assumption of a straight-line relationship but are smoothers from a GAMM instead; both axes are on a log scale; blue "violins": mirror density plots with horizontal lines representing the median, upper and lower quartile, red line: a smoother for the overall relationship (predicted fixed effects from the model), grey line: a smoother for each individual antlion (predicted random effects from the model).
(PDF)

**S2 Fig. Antlion movement duration (s) against the sequential number of the movement period since predator contact for each of the three substrates: Paper, Shallow sand, Deep sand.** It is the same as Fig 2 except that the fitted lines are not based on the assumption of a straight-line relationship but are smoothers from a GAMM instead; both axes are on a log scale; blue "violins": mirror density plots with horizontal lines representing the median, upper and lower quartile, red line: a smoother for the overall relationship (predicted fixed effects from the model), grey line: a smoother for each individual antlion (predicted random effects from the model, very similar within each treatment here).
(PDF)

**S3 Fig. Individual paths with stops and displacements between stops on the three substrates: Path (dotted black line) from the spot in the arena centre where the antlion larva was dropped (Paper: Light blue circle; Shallow sand: Light green circle; Deep sand: Light yellow circle) to the wall (Paper: Red circle) or end of at most 16 moves (Shallow sand: Pink circle; Deep sand: Salmon pink circle) with superimposed stops (Paper: Dark blue circle; Shallow sand: Dark green circle; Deep sand: Dark yellow circle) and displacement segments between stops (Paper: Light blue line; Shallow sand: Light green line; Deep sand: Light yellow line); all antlions tested on Shallow or Deep sand were also tested on Paper and the number for each substrate are: N = 22 (Paper), N = 8 (Shallow sand), N = 8 (Deep sand); note that antlions 10 and 13 made only two moves and two stops and did not reach the arena wall within the ~90 min of observation on Paper.**
(PDF)

**S4 Fig. Individual paths with stops and displacements between stops on Shallow sand: Path (dotted black line) from the spot in the arena centre where the antlion larva was dropped (light green circle) to the end of 16 moves (pink circle) with superimposed stops (dark green circle) and displacement segments between stops (light green line); these are the same paths as in S3 Fig but on a larger scale.**
(PDF)

**S5 Fig. Individual paths with stops and displacements between stops on Deep sand: Path (dotted black line) from the spot in the arena centre where the antlion larva was dropped (light yellow circle) to the end of 16 moves (salmon pink circle) with superimposed stops (dark yellow circle) and displacement segments between stops (light yellow line); these are the same paths as in S3 Fig but on a larger scale.**
(PDF)

**S6 Fig. Individual instantaneous speed (mms$^{-1}$) over time (s) on Paper: Instantaneous speed over time (black line) calculated from the image-analysis track and the superimposed observed movement periods (cyan highlight); note that antlions 10 and 13 made only two moves and two stops and did not reach the arena wall within the ~90 min of**

observation.
(PDF)

**S7 Fig. Individual instantaneous speed (mms$^{-1}$) over time (s) on Shallow sand: Instantaneous speed over time (black line) calculated from the image-analysis track and the superimposed observed movement periods up to 16 moves (cyan highlight).**
(PDF)

**S8 Fig. Individual instantaneous speed (mms$^{-1}$) over time (s) on Deep sand: Instantaneous speed over time (black line) calculated from the image-analysis track and the superimposed observed movement periods up to 16 moves (cyan highlight).**
(PDF)

**S1 Data. Data dictionary for S1 Table.**
(XLSX)

**S2 Data. R scripts for statistical analyses.**
(ZIP)

**S3 Data. Data for antlion tracks on paper.**
(ZIP)

**S4 Data. Data for antlion tracks on shallow sand.**
(ZIP)

**S5 Data. Data for antlion tracks on deep sand.**
(ZIP)

**S6 Data. Data for antlion start-to-finish displacement.**
(ZIP)

## Acknowledgments

We thank our Departments for their support.

## Author Contributions

**Conceptualization:** Nigel R. Franks, Alan Worley, George T. Fortune, Ana B. Sendova-Franks.

**Data curation:** Alan Worley, George T. Fortune, Ana B. Sendova-Franks.

**Formal analysis:** Alan Worley, George T. Fortune, Ana B. Sendova-Franks.

**Funding acquisition:** George T. Fortune, Raymond E. Goldstein.

**Investigation:** Nigel R. Franks, Alan Worley, George T. Fortune, Ana B. Sendova-Franks.

**Methodology:** Nigel R. Franks, Alan Worley, George T. Fortune, Raymond E. Goldstein, Ana B. Sendova-Franks.

**Project administration:** Nigel R. Franks, Ana B. Sendova-Franks.

**Resources:** Nigel R. Franks, Raymond E. Goldstein.

**Software:** Alan Worley, George T. Fortune.

**Supervision:** Raymond E. Goldstein.

**Validation:** Nigel R. Franks, Alan Worley, George T. Fortune, Raymond E. Goldstein, Ana B. Sendova-Franks.

**Visualization:** Nigel R. Franks, Alan Worley, George T. Fortune, Raymond E. Goldstein, Ana B. Sendova-Franks.

**Writing – original draft:** Nigel R. Franks, Ana B. Sendova-Franks.

**Writing – review & editing:** Nigel R. Franks, Alan Worley, George T. Fortune, Raymond E. Goldstein, Ana B. Sendova-Franks.

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
