## [Decision Letter · Decision Letter 0]

12 Dec 2023

PONE-D-23-29142Seeking safety: movement dynamics after post-contact immobilityPLOS ONE

Dear Dr. Franks,

Thank you for submitting your manuscript to PLOS ONE. After careful consideration, we feel that it has merit but does not fully meet PLOS ONE’s publication criteria as it currently stands. Therefore, we invite you to submit a revised version of the manuscript that addresses the points raised during the review process.

I would like to sincerely apologise for the delay you have incurred with your submission. It has been exceptionally difficult to secure reviewers to evaluate your study. We have now received four completed reviews; the comments are available below. The reviewers have raised significant scientific concerns about the study that need to be addressed in a revision.

Please revise the manuscript to address all the reviewer's comments in a point-by-point response in order to ensure it is meeting the journal's publication criteria. Please note that the revised manuscript will need to undergo further review, we thus cannot at this point anticipate the outcome of the evaluation process.

We look forward to receiving your revised manuscript.

Kind regards,

Miquel Vall-llosera Camps

Staff Editor

PLOS ONE

Journal Requirements:

"This work was supported in part by the Engineering and Physical Sciences Research Council through a doctoral training fellowship (G.T.F.) and the Schlumberger Chair Fund at DAMTP, University of Cambridge (R.E.G.)."

Additional Editor Comments:

We note that one or more reviewers has recommended that you cite specific previously published works. As always, we recommend that you please review and evaluate the requested works to determine whether they are relevant and should be cited. It is not a requirement to cite these works. We appreciate your attention to this request.

Reviewers' comments:

Reviewer's Responses to Questions

**Comments to the Author**

1. Is the manuscript technically sound, and do the data support the conclusions?

Reviewer #1: Partly

Reviewer #2: Yes

Reviewer #3: Partly

Reviewer #4: Yes

2. Has the statistical analysis been performed appropriately and rigorously? 

Reviewer #1: Yes

Reviewer #2: Yes

Reviewer #3: I Don't Know

Reviewer #4: Yes

3. Have the authors made all data underlying the findings in their manuscript fully available?

Reviewer #1: Yes

Reviewer #2: Yes

Reviewer #3: Yes

Reviewer #4: Yes

4. Is the manuscript presented in an intelligible fashion and written in standard English?

Reviewer #1: Yes

Reviewer #2: Yes

Reviewer #3: Yes

Reviewer #4: Yes

5. Review Comments to the Author

Reviewer #1: Dear authors

I have read your study entitled »Seeking saftey: movement dynamics after post-contact immobility«. It is an interesting study with general appeal, and knowing your previous studies, it provides additional information on how antlions respond to a putative predator after post-contact immobility. Your statistical methods are quite extensive. You have taken a very quantitative approach to your study. I think this is good, but to balance this we also need a simple description of behaviour that best illustrates the actual state of the animal under and after PCI. Measuring behaviour using mostly a statistical approach is not enough. This is only consideration. The description of behaviour and, in addition, physiological measurements can provide important information about whether the animals are under stress or not. The biological background of these animals is also important. E. nostras is a habitat specialist that lives in sheltered places and avoids direct sunlight. Some conditions in your study may have affected your measurements, especially given the small number of animals studied. In some parts of the discussion you make comparisons with other studies, which I think is hardly possible based on the results of your study.In general I like your study, but biological background conected to animals you tested is somewhat missing.

I have some minor comments on some parts of the manuscript that I think need to be explained:

Keywords: you use different keywords for the same thing. You have particularly focused on mobility after PCI.

I have some minor comments on some part of the masucript which I think needs to be explained:

Key words: you use different key words for the same thing. You focused especially on mobility after post-contact. Mybe you consider deleting some and add »antlion larvae«, »sit-and-wait animals« as this could be important for other studies that you elucidate you had sit-and-wait animals.

Line 57: mybe you add (in our case antlions)

Line 79: »can span more than three orders of magnitude« why not just write »can last«

Lines 82-86: you explain some statistical data acording to previous studies. They are not necessary. Just continue with line 87.

Line 91: delete »an endless future«

Line 125: mybe you should add »sit-and-wait organisms« as this could be important in comparison of PCI and movements after that in comparison to other more active animals.

Line 131: it is not an option it is innate fixed-action pattern when antion are on surface. Consider rewriting sentence.

Line 136: is 4 mm really deep sand? You refer further that 2 and 4 cm was used in previous studies, which is definetely more appropriate for deep and shallow sand. And as you explain further in such sand the pit-building is not possible. True, but also submergence in 4 mm sand is not quite possible for the third instar (larger larvae). The larva hits the hard surface beneth before it is fully submerged.

Line: what is fine-tuned ability? Please explain.

Line 162: what do you mean »two sand substrates«? You mean two sand depth? If you refer to »two subsrates« it can be understand you had different coarsesnes.

Line 161-164: it is not very clear what you are trying to formulate. The larvae have innate behaviour to burry under the substrate after they find themself on the surface. Untill they start burrying themself they can not percive the difference in sand depth – so this question is not understandable. Moreover, you had different larval stages and they probably experienced sand depth in different ways.

. Maybe you should delete some and add »antlion larvae«, »sit-and-wait animals« as this might be important for other studies

Line 57: maybe add (in our case antlions)

Line 79: »can span more than three orders of magnitude«, why not just write »can take/last« or something similar.

Lines 82-86: you explain some statistical data based on previous studies. They are not necessary. Just continue with line 87.

Line 91: delete »an endless future«

Line 125: perhaps you should add »sedentary organisms« as this might be important for comparing PCI and subsequent movements compared to other, more active animals.

Line 131: it is not an option but an innate behavioural pattern when the antions are on the surface. Perhaps you should rewrite the sentence.

Line 136: is 4 mm really deep sand? You point out that previous studies have used 2 and 4 cm, which is definitely more appropriate for deep and shallow sand. Is complete submersion in 4 mm deep sand possible for the third instar (larger larvae)? The larva hits the hard surface before it is fully submerged.

Line: What is a fine-tuned ability? Please explain.

Line 162: What do you mean by »two substrates of sand«? Do you mean two depths of sand? If you are referring to »two substrates«, it can be assumed that you had different coarsenesses.

Lines 161-164: It is not entirely clear what you are trying to formulate. The larvae have an innate behaviour to bury themselves under the substrate after they are on the surface. As long as they do not bury themselves, they cannot perceive the difference in sand depth – so this question is not understandable. Also, you had different larval stages that probably perceived the sand depth in different ways.

Line 170: »most larval stages« define what larval stages you had. With the measurements of the head and body this is easy, but according to your mass you probably had the 2nd and 3rd instar. It might be difficult to manipulate with the first instar – also, it's probably difficult to compare the duration of movements in such different sized animals. Probably the first or second instar needs greater challenges to move through the substrate and walks less and stops more often. It is very speculative to observe and define behaviour in such a small number of test animals and such a variety of stages. Just one consideration.

Line 173: Do you provide water to the larvae othervise? And how? What temperature were the larvae at when they were isolated? This could be important as temperature could significantly affect the duration of PCI as antlions are ectothermic organisms. Were the larvaein vials with sand or without sand when they were isolated? Please provide information.

Line 174: perhaps it is more important that E. nostras is a habitat specialist that chooses places protected from rain and direct sunlight.

Lines 184-186: you stated that they spend most of their lives in darkness, but in the experiment you placed lights over the arenas. Do you think this was not disruptive? And could it have an influence on the PCI and the behaviour of the animals after the PCI?

Line 187: You should specify what type of substrate particles you used for the study. Since you had larvae of different sizes, this is very important.

Line 228: You wrote that you observed the antlions for 90 minutes. Why did you stop after the antlion had reached the arena wall? Or have I misunderstood? It would be interesting to know how the antlions moved after hitting the arena wall. Did they return to the centre of the petri dish or did they just move around the edge. This is an important anti-predator behaviour that should be observed, especially in cases where burying is not possible.

Line 264: Rather use "toss" or "throw" instead of "kick"

Kind regards.

Reviewer #2: I read the manuscript by Nigel Franks et al. with great interest. The authors present a comprehensive study of post-contact immobility in predatory behaviour of antlions. The study is a very interesting paper, containing valuable new information, worthy of publication. I suggest to accept it after minor revision.

Comments and suggestions:

Lines 72-74:

Antlions are also prey by conspecifics. It would be wise to mention immobility in such cases. See also a paper by Klokocovnik et al. 2020:

Klokočovnik V., Veler E., Devetak D. 2020. Antlions in interaction: confrontation of two competitors in limited space. Israel Journal of Ecology & Evolution, 66 (1-2): 73-81.

DOI: 10.1163/22244662-20191058

Lines 152-153:

"Hard or soft substrates" - These terms refer to density. The following paper could be cited here: Devetak D., Novak T., Janžekovič F. 2012: Effect of substrate density on behaviour of antlion larvae (Neuroptera: Myrmeleontidae). Acta Oecologica, 43: 1–7.

http://dx.doi.org/10.1016/j.actao.2012.05.010

Lines 610-612:

Here you should cite also a review paper dealing with the role of the sensory hairs in antlions:

Devetak, D. 2014. Sand-borne vibrations in prey detection and orientation of antlions. In: R.B. Cocroft, M. Gogala, P. S. M. Hill, A. Wessel (eds.): Studying Vibrational Communication. Animal Signals and Communication 3, DOI: 10.1007/978-3-662-43607-3_16. Berlin, Heidelberg: Springer-Verlag, pp. 319-330.

Reviewer #3: Dear Authors,

I have reviewed the manuscript “Seeking safety: movement dynamics after post-contact immobility” by Franks et al. for possible publication in journal PLOS ONE. Authors have investigated the post contact mobility/immobility after in antlion larvae after post contact immobility induced by authors. The article is well written, a bit lengthy in certain parts. The topic is interesting as post contact immobility is an important behavioural pattern, which is still not sufficiently explored, especially what happens after PCI.

The topic and the concept of the experiment is very interesting. The quantitative approach used for analysis of this phenomena has quite some potential. However, there are some major issues regarding the methodology and experimental procedure.

One problem is the temperature range in which the experiment has been executed. There are several studies that are concerned with the impact of temperature on physiology and behaviour of insects, even on antlion larvae. Five degree (5 °C) difference could influence the duration of immobility/mobility between replicates. Another problematic aspect of this study is the sample size (see the remarks below).

Comparing movement on hard surface and in the sand is quite challenging task in antlion larvae, as antlions use one substrate for burrowing and hiding (their natural microhabitat), while hard surfaces are inappropriate substrate for sand dwelling species and it rushes to find cover as soon as possible. This is reflected in the results of this study. We do not know if the results would differ if antlion larvae would be placed gently on the surface instead of being dropped. So the question is, are we observing the effect of PCI induced by different substrates (hardness) or simply the effect of different substrate on antlion larva, which could be the same even if the animal is not dropped on the surface.

Unless authors are able to properly address these issues I do not think that the manuscript is acceptable for publication in PLOS ONE.

Other remarks

L55: Consider adding the term thanatosis, which is widely used.

L134-151: Part of this paragraph should probably be in the method section. Retain only enough information for the purpose/goal of the experiment remains clear.

L167: Small sample size, probably too small (especially for both sand experiments).

L169-171: There is quite a range in animal weight. You probably used different instars. You should provide information: which instars were used in the experiment and number of individuals from each instar. There is evidence about difference in behaviour/activity of different instars (Alcalay et al. 2014 – article cited below). Did you notice any difference in PCI and post PCI behaviour in regard to the size of animals?

Additionally: you give the mass of median, LQ and UQ with more precision than your measurements allow. Please correct it.

Alcalay Y, Barkae ED, Ovadia O, Scharf I. Consequences of the instar stage for behavior in a pit-building antlion. Behav Processes. 2014 Mar;103:105-11. doi: 10.1016/j.beproc.2013.11.009. Epub 2013 Dec 4. PMID: 24315799.

L171-172: Temperature variation was 5 °C, which is high, especially when analysing parameters like immobility and mobility duration. There are several papers on this topic.

L188-192: What was the substrate particle size?

L191-193: This an unusual formulation. A “typical antlion larva”? I believe there is no such thing. There are 3 instars of antlion larvae. They differ substantially in size. You should rephrase.

L193-194: For a small first instar larva the 2 mm of sand should be enough to cover/burry itself with sand. Therefore, it is important, as stated before, that you provide information about the instar of your individuals and number of individuals from each instar.

L212-216: Even though, authors justify the small sample size (especially for the sand substrate experiment) with experimental design and use of appropriate statistical method, I question if 8 antlion larvae (even 5 in one case) of different mass and larval stage can be representative sample for making strong conclusions.

L285: As pointed out before, the sample size is important for drawing conclusions from obtained results. I suggest additional reviewer skilled/competent in statistical methodology.

Reviewer #4: The authors examine the pattern of movement and immobility when antlion larvae resume activity after Post-Contact immobility (PCI). To simulate contact with, and escape from a predator they dropped the larvae onto three substrates: paper, shallow sand, and deep sand. The authors tracked the antlion's paths automatically and recorded alternating immobility and movement durations over 90 minutes. Their results suggest that post-PCI movement and immobility gradually return to the pattern typical of intermittent locomotion, depending on the scope for burying and hiding in the substrate.

I found the topic of this manuscript interesting. The work also shows an adequate experimental design (despite the low number of replicates and the use of paper instead of other more natural substrates, see below), and is relatively easy to read (despite the great relevance given to mathematics and statistics instead of biology, see below). I have only a few comments that I hope help to improve this nice manuscript.

First, my feeling is that the authors try to calculate the area of a square by integrating instead of multiplying their sides. This metaphor intends to explain my sensation about the use of complex statistical and math tools to demonstrate what seems a predictable and simple behavior: after PCI, in hard and shallow substrates, antlions move to look for a better place to dig, but in deep sand, antlions move mainly to buried themselves. This does not invalidate the interesting and novel topic of the manuscript, but –in my opinion, the authors take a complicated way instead of a simple one to reach a biological conclusion that is relatively predictable.

Second, I think that dropping antlion on a stone surface would have been a much better –and real- option than dropping them on a paper surface. I understand that this experimental design can no longer be modified, but the use of paper instead of stone should be better justified.

Third, the authors predict that antlions will search for the relative safety of escaping altogether from a hard substrate since they cannot hide by digging. I agree, but ¿there are other options? In other words, if escape from this substrate is the only alternative, this result does not necessarily support the hypothesis that generates it. I believe that it is necessary to clarify if different alternatives exist and why escape is the one that is an unequivocal consequence of the hypothesis.

L. 126. Farji-Brener (2003) could be an adequate reference here.

Reference

Farji-Brener, A. G. (2003). Microhabitat selection by antlion larvae, Myrmeleon crudelis: effect of soil particle size on pit-trap design and prey capture. Journal of Insect Behavior, 16(6), 783-796.

6. PLOS authors have the option to publish the peer review history of their article (what does this mean?). If published, this will include your full peer review and any attached files.

Reviewer #1: No

Reviewer #2: No

Reviewer #3: No

Reviewer #4: No

---

## [Author Response · Author response to Decision Letter 0]

8 Jan 2024

Please see the uploaded file "Response to Reviewers PONE-D-23-29142".

---

## [Decision Letter · Decision Letter 1]

13 Feb 2024

PONE-D-23-29142R1Seeking safety: movement dynamics after post-contact immobilityPLOS ONE

Dear Dr. Franks,

Thank you for submitting your manuscript to PLOS ONE. After careful consideration, we feel that it has merit but does not fully meet PLOS ONE’s publication criteria as it currently stands. Therefore, we invite you to submit a revised version of the manuscript that addresses the points raised during the review process.

We look forward to receiving your revised manuscript.

Kind regards,

Daniel de Paiva Silva, Ph.D.

Academic Editor

PLOS ONE

Additional Editor Comments:

Dear Dr. Franks,

After this new review round, both reviewers recognize you have reached a significant improvement in your manuscript, from the original version to the newly submitted version. One of the reviewers is satisfied with the new version and approved its acceptance. Still, the other reviewer is still concerned about the overall design of the experiment. The first issue regards the temperature range used in the experiment. According to their view, a 5-degree range significantly affects the insect's behavioral repertory, which I also tend to agree with. Another issue is that the reviewer believes comparing Antlion's movements on both hard/sand surfaces is also troublesome and should affect the experiment's results, which I also think is an important issue that may affect the obtained results. Finally, other minor improvements are suggested and (as far as I can see) should be addressed also in the next version of the manuscript.

In my experience as an editor and working with insect behavior a long time ago, I consider you would have two paths to solve the issues raised by the reviewer: one that may seem more "easy" but that may prove to be fatal and a more difficult one. The "easy" one would be to add information on how antlion experiments similar to yours are made, explaining how the general research with these insects in your field of research is made and how your experimental design complies with the other studies, showing there was no fatal flaw with the methods you employed. The second and more difficult one would be to set up a new experiment, showing if different manipulation strategies and temperature ranges do not affect the results you obtained. If the result of the experiment considering different manipulation strategies and temperature ranges does not show any significant difference among the considered variables, I believe the reviewer will (hopefully) provide their blessings to your manuscript and accept it for publication. I want to stress that this is also possible by following the "easy" path. Still, please note that in such a path, in the end, the reviewer may not agree with what has been argued and may end up rejecting the manuscript.

Since this is not an easy choice, I will provide you with a four-month period (June 15th, 2024) to deliver the new version of the manuscript. If you decide to follow the more difficult path, I believe you will have enough time (I think) to plan this new experiment. Still, in case you decide on a difficult path and still need more time, please let me know, and I will grant you more time to complete the decided task. Finally, I would like to acknowledge the efforts made by the reviewers and I hope to count on their knowledge in the next version of the study.

Sincerely,

Daniel Silva, PhD

Reviewers' comments:

Reviewer's Responses to Questions

**Comments to the Author**

1. If the authors have adequately addressed your comments raised in a previous round of review and you feel that this manuscript is now acceptable for publication, you may indicate that here to bypass the “Comments to the Author” section, enter your conflict of interest statement in the “Confidential to Editor” section, and submit your "Accept" recommendation.

Reviewer #3: All comments have been addressed

Reviewer #4: All comments have been addressed

2. Is the manuscript technically sound, and do the data support the conclusions?

Reviewer #3: Partly

Reviewer #4: Yes

3. Has the statistical analysis been performed appropriately and rigorously? 

Reviewer #3: Yes

Reviewer #4: Yes

4. Have the authors made all data underlying the findings in their manuscript fully available?

Reviewer #3: Yes

Reviewer #4: Yes

5. Is the manuscript presented in an intelligible fashion and written in standard English?

Reviewer #3: Yes

Reviewer #4: Yes

6. Review Comments to the Author

Reviewer #3: Dear Authors,

I have read the revised manuscript “Seeking safety: movement dynamics after post-contact immobility” by Franks et al.

Thank you for extensive clarifications of comments given in previous round of review. Most clarifications are satisfactory. I still believe that the manuscript has some merit, especially in methodological approach. However, there is still one issue I am concerned with. Same issues were raised by other reviewers as well (except one) and your answer did not satisfy me.

Authors explore what is the course of events in a potential prey after it was stressed, dropped on the surface, which induced a thanatosis state, a behavioural mechanism for avoiding predators when hiding is not possible. While dropping animals on different substrates induces PCI of different duration (not statistically significant in this study), what happens after this behavioural pattern ends, is also a function of very simple reflex behaviour of an animal that submerges into substrate when this is possible and seeks a place to submerge when this stereotypic behaviour is not an option. This means that it is not clear if the induction of PCI has a crucial contribution to events that happen after it – the events this study is exploring. It could be a normal response of an antlion larva to the different substrate regardless of before induced PCI. The experiment is lacking a control, which would validate this part of author’s hypothesis.

The second problem I see is the comparison of movement on hard surface and in the sand. Antlions use one substrate for burrowing and hiding (their natural microhabitat), while hard surfaces are inappropriate substrate for sand dwelling species, and it rushes to find cover as soon as possible. This is reflected in the results of this study – predictable outcome, as hard surface does not offer anything else than looking for place to bury itself. You say: “The question is how they balance the immediate danger from the putative predator that has dropped them against the future danger of being exposed on the substrate surface. “ So the answer is: They burry them self if possible – sand substrate -, if not, they use a certain pattern of immobility and mobility, which is an answer to the trade-off you mention. But you do not know if this pattern is any different if no PCI is induced before.

Because of these conceptual issues I still have concerns about the manuscript suitability for publication in PLOS ONE Journal.

Minor comments:

L134-151: I still believe that a big part of this paragraph belongs in the method section. Retain only enough information for the purpose/goal of the experiment remains clear or it becomes even clearer.

Reviewer #4: I just read the second version of the paper entitled “Seeking safety: movement dynamics after post-contact immobility”

In general, I found this version clearer than the previous one. I believe that this interesting paper is ready to be published. I only have minor comments regarding my previous suggestions.

First, I still feel that the use of complex statistical and math tools to demonstrate what seems like a predictable and simple behavior is not the best way. I am glad that the authors follow my metaphor about how is the better and simpler way to calculate the area of a square, and I fully understand that “integrals are more important than areas because areas are crucial to geometry, while integrals are crucial to everything” But it is also true that simpler analyses are “easier to explain and understand; they clarify what the key units in a study are; they reduce the chances for computational mistakes; and they are more likely to lead to the same conclusions when applied by different analysts to the same data” (Murtaugh 2007). Maybe is an interesting topic to discuss in other contexts (e.g.; drinking a beer) rather than here. Second, I also understand that the paper lining the bottom of a plastic Petri dish represents a hard impenetrable surface, but this scenario is not the best representation of what happens in nature. The authors could discuss and make clear what real scenario this lab condition tries to represent (e.g. when larvae are drooping on compact soil, a stone surface, or on a fallen trunk) and argue their eventual limitations. Finally, if the substrate is so hard that impedes the larvae buried themselves, there are no other expected results than keeping immobile or moving looking for a better place to be buried. I believe that it is necessary to clarify if different alternatives exist and why escape is the one that is an unmistakable consequence of the hypothesis.

Congratulations for your nice work!

References

Murtaugh, P. A. (2007). Simplicity and complexity in ecological data analysis. Ecology, 88(1), 56-62.

7. PLOS authors have the option to publish the peer review history of their article (what does this mean?). If published, this will include your full peer review and any attached files.

Reviewer #3: No

Reviewer #4: No

---

## [Author Response · Author response to Decision Letter 1]

21 Feb 2024

Please see the uploaded document "Response to Reviewers PONE-D-23-29142R1".

---

## [Decision Letter · Decision Letter 2]

16 Apr 2024

PONE-D-23-29142R2Seeking safety: movement dynamics after post-contact immobilityPLOS ONE

Dear Dr. Franks,

Thank you for submitting your manuscript to PLOS ONE. After careful consideration, we feel that it has merit but does not fully meet PLOS ONE’s publication criteria as it currently stands. Therefore, we invite you to submit a revised version of the manuscript that addresses the points raised during the review process.

We look forward to receiving your revised manuscript.

Kind regards,

Daniel de Paiva Silva, Ph.D.

Academic Editor

PLOS ONE

Additional Editor Comments:

Dear Dr. Franks,

After this new review round, a new reviewer still raised significant issues that need to be considered in the new version of your text. The reviewer that accepted the MS made similar reservations regarding the text to me, letting me decide the fate of your manuscript. Since I am not an expert in the topics covered in your text, and since two experts reached the same decision, although one let me decide if I should accept it or not, I believe the correct decision to make is to be able to convince reviewer #5. See, the amount of issues raised is considerable and, at this point, I believe I could only accept the text to be published if, in the next version, reviewer #5 believes the text deserves minor reviews from there on. Therefore, I will grant you a new 2-month period to deliver your manuscript and rebuttal letter.

Sincerely,

Daniel Silva

Reviewers' comments:

Reviewer's Responses to Questions

**Comments to the Author**

1. If the authors have adequately addressed your comments raised in a previous round of review and you feel that this manuscript is now acceptable for publication, you may indicate that here to bypass the “Comments to the Author” section, enter your conflict of interest statement in the “Confidential to Editor” section, and submit your "Accept" recommendation.

Reviewer #1: (No Response)

Reviewer #5: (No Response)

2. Is the manuscript technically sound, and do the data support the conclusions?

Reviewer #1: Yes

Reviewer #5: Partly

3. Has the statistical analysis been performed appropriately and rigorously? 

Reviewer #1: Yes

Reviewer #5: I Don't Know

4. Have the authors made all data underlying the findings in their manuscript fully available?

Reviewer #1: Yes

Reviewer #5: No

5. Is the manuscript presented in an intelligible fashion and written in standard English?

Reviewer #1: Yes

Reviewer #5: Yes

6. Review Comments to the Author

Reviewer #1: (No Response)

Reviewer #5: Dear Authors,

I have now read the manuscript titled “Seeking safety: movement dynamics after post-contact immobility” (PONE-D-23-29142R2). In this study the authors investigated the defensive behavior of an antlion after being dropped by a surrogate predator on different surfaces (deep sand, shallow sand and paper). The paper is overall well written, and the research questions are relevant. Moreover, the authors advocate the importance of using quantitative approaches in post-contact immobility (AKA tonic immobility, death feigning, thanatosis, to name a synonymous). This suggestion is among the highlights of this manuscript, and the literature on the topic would greatly benefit from better understanding the quantitative properties of this behavior across animal taxa. Even though the manuscript is of good quality, I have a few major concerns that are listed below (by section) followed by minor comments/suggestions:

Major comments

Experimental design:

- While the treatment and controls chosen are appropriate to answer the chosen biological question, I am afraid that a few aspects of the current design and their possible effects on the results deserve further attention:

1-The authors did not standardize the feeding state of the experimental models: It is well documented in the scientific literature that the level of starvation can affect the behavior of arthropods. Please see Scharf (2016). Anim Behav, 119, 37-48 and references therein. As such, one cannot rule out the fact that by chance individuals bearing different levels of starvation may have been disproportionally chosen for different experimental groups. Especially given the relatively low sample size used in this study (see topic below). Therefore, the results may have been confounded by the non-standardized body state of the experimental models.

2-The samples size are strongly unbalanced. While the experimental group tested with paper subtract had a relatively acceptable sample size (N = 22), for the both the groups tested with subtracts Deep and Shallow, one could argue that the sample size was too low (N = 8), especially considering the high variance typically found in behavioral studies. With that being said, I am afraid that this unbalance led to an augmented chances of incurring in an error type I [see Forstmeier et al (2017). Biol Rev 92,1941-1968]. The authors made use of bootstrapping to deal with this issue, but resampling the measures may not be enough if the collected data does not provide correct estimates of the underlying population.

3-Given that the group “paper substrate” had a significantly larger sample size than the other two “Deep and Shallow Sand”: I wonder if the time of the test might not have influenced disproportionally the experimental groups? Additionally, given the repeated measures experiment, I would request the authors to provide a file with the order of the tests (a sequence informing the order in which each sampled individual went through concerning the experimental groups and time of the day when they were tested)

Discussion:

The discussion section of the manuscript would greatly benefit from making comparisons of the findings of their study with similar results from the literature showing defensive behavior to be context dependent [see also Threat Sensitive Hypothesis, e.g. Gyssels & Stoks (2005). Ethology, 111: 411-423], especially in “post-contact immobility” for example, concerning daily cycle [Jones et al (2011). Anim Behav 82:549–555; Miyatake (2001). J Insect Behav 14:421–432; Segovia et al (2019). J Arachnol, 47:396-398; Watts et al (2014). Anim Behav 94:79– 86].

I struggle with the idea of assuming that a pattern of behavior is evidence of distress on its own. When a given behavior is associated with an objective mechanistic measure of stress (e.g. cortisol levels) in the very same species, one might speculate this behavior to be a proxy of distress yet incurring in some risk. However, if that is not the case, one could call whatever different behavioral pattern emerges a measure of distress, and I suspect that it is not the most parsimonious approach. As such, I would suggest the authors to either rethink the use of this term in the manuscript or build a more solid argument on why that behavior is a valid measure of distress (if that is the case).

Additionally, it would be desirable to include a paragraph in the discussion section about the possible limitations resulting from the experimental design (Comments 1-3, in experimental design section above)

Minor comments:

Theoretical approach:

L 177: Please provide metadata for S1Table.

L 186-187: Were the effects of time and temperature added as random effects in the models? If possible, the readers could benefit from having access to the scripts with the fitted models.

L 191-193: Please clarify.

L 201: Have you used the same sand described above?

L 202: Please provide the data (a measure of central tendency and deviation) concerning the size of the experimental models, the best case would be if you could provide it by experimental groups.

L 205-208: Could you please clarify it? It would be done even by rephrasing or providing a schematic picture.

L 215-207: Shall you please be more specific about the criteria adopted to distribute the larvae according to the weight (experimental groups? Time?).

L 231-235: If the order of the treatments was not randomized, please provide precise information on how it was done, and add to the discussion possible implication of the non-randomized order on your results.

L 233-235: The fact that the behavior is repeatable does not rule out the possibility of an interaction with the treatment in many different directions, please acknowledge this in the manuscript.

L 239-240: What was done with the measurements from these individuals?

L 245-249: Please delete from L 245, merge or rephrase.

L 254-256: How were they defined? For instance, have you considered movements of non-locomotory appendage as movement or not? Please clarify.

L293-296: Shall it not be possible that the antlions moved and returned to a shorter-range distance in between the initial (after dropping) and final measurements? If that is the case, please discuss this possibility in the manuscript.

L 305-306: The use of the word movement withing brackets twice in this sentence (for immobility duration and successive immobility), seems to be somewhat misleading, may you please clarify it?

L 325: Which sample size you are mentioning to be large? Please specify.

L 559: shouldn't the first movement after the PCI be expected to be the fastest? In case they noticed right after falling that they cannot burrow in that surface.

L 638: The argument that you may have not found the expected differences due to the small sample size can be extended to the risk of incurring in type 1 error, and as such it is desirable to add a paragraph about the limitations of the findings of this study.

L 640: see also Segovia et al (2019). Curr Zool 65: 553-558.

L 643-645: Very important suggestion!

Table 2: Maybe the lack of difference in the slope between Shallow and Deep Sand is related to the small sample size of these groups? Shall it not be worthy of a discussion?

7. PLOS authors have the option to publish the peer review history of their article (what does this mean?). If published, this will include your full peer review and any attached files.

Reviewer #1: No

Reviewer #5: No

---

## [Author Response · Author response to Decision Letter 2]

14 May 2024

Please see the uploaded file "Response to Reviewers PONE-D-23-29142R2.docx".

---

## [Decision Letter · Decision Letter 3]

27 May 2024

PONE-D-23-29142R3Seeking safety: movement dynamics after post-contact immobilityPLOS ONE

Dear Dr. Franks,

Thank you for submitting your manuscript to PLOS ONE. After careful consideration, we feel that it has merit but does not fully meet PLOS ONE’s publication criteria as it currently stands. Therefore, we invite you to submit a revised version of the manuscript that addresses the points raised during the review process.

We look forward to receiving your revised manuscript.

Kind regards,

Daniel de Paiva Silva, Ph.D.

Academic Editor

PLOS ONE

Journal Requirements:

Additional Editor Comments:

Dear Dr. Franks,

We are almost there! After this new review round the remaining reviewer asked for very minor changes to be made and as soon as the new version is submitted, I beleive the manuscript will be accepted.

Sincerely,

Daniel Silva

Reviewers' comments:

Reviewer's Responses to Questions

**Comments to the Author**

1. If the authors have adequately addressed your comments raised in a previous round of review and you feel that this manuscript is now acceptable for publication, you may indicate that here to bypass the “Comments to the Author” section, enter your conflict of interest statement in the “Confidential to Editor” section, and submit your "Accept" recommendation.

Reviewer #5: All comments have been addressed

2. Is the manuscript technically sound, and do the data support the conclusions?

Reviewer #5: Partly

3. Has the statistical analysis been performed appropriately and rigorously? 

Reviewer #5: Yes

4. Have the authors made all data underlying the findings in their manuscript fully available?

Reviewer #5: No

5. Is the manuscript presented in an intelligible fashion and written in standard English?

Reviewer #5: Yes

6. Review Comments to the Author

Reviewer #5: The authors carefully answered my comments, and added two paragraphs in the discussion section, as well as multiple fragments of information across the text. The information included in the revised version improved the clarity of the manuscript concerning the methods used and its limitations. In my opinion, the manuscript has improved a lot and is now ready for publication. However, I still have two minor comments that I suggest the authors to consider:

1- Adding metadata to a dataset improves its readability, as well as the reproducibility of the results. As such I would recommend the authors to look at the link (https://the-turing-way.netlify.app/reproducible-research/rdm/rdm-metadata#rr-rdm-metadata) and consider to add a data dictionary (https://help.osf.io/article/217-how-to-make-a-data-dictionary)

2- For the comment/answer discussed in the last round below:

“23. L 325: Which sample size you are mentioning to be large? Please specify.

We are referring to the sample sizes for the stated models. They are large because the modelled data include between 2 and 16 repeated measures for each antlion on each of three substrates. I would suggest the authors to add this information in the text, because I was surprised to read it at first, as it seemed to me that the authors were mentioning the actual sample size of the experiment. Adding a sentence could help the reader to not be misguided.

7. PLOS authors have the option to publish the peer review history of their article (what does this mean?). If published, this will include your full peer review and any attached files.

Reviewer #5: No

---

## [Author Response · Author response to Decision Letter 3]

23 Jun 2024

Please see the uploaded file "Response to Reviewers PONE-D-23-29142R3.docx".

---

## [Decision Letter · Decision Letter 4]

4 Jul 2024

Seeking safety: movement dynamics after post-contact immobility

PONE-D-23-29142R4

Dear Dr. Franks,

We’re pleased to inform you that your manuscript has been judged scientifically suitable for publication and will be formally accepted for publication once it meets all outstanding technical requirements.

Kind regards,

Daniel de Paiva Silva, Ph.D.

Academic Editor

PLOS ONE

Additional Editor Comments (optional):

Dear Dr. Franks,

THe hardwork always pay off! I am pleased to accept your manuscript for publication in PLoS One.

Sincerely,

Daniel Silva

Reviewers' comments:

Reviewer's Responses to Questions

**Comments to the Author**

1. If the authors have adequately addressed your comments raised in a previous round of review and you feel that this manuscript is now acceptable for publication, you may indicate that here to bypass the “Comments to the Author” section, enter your conflict of interest statement in the “Confidential to Editor” section, and submit your "Accept" recommendation.

Reviewer #5: All comments have been addressed

2. Is the manuscript technically sound, and do the data support the conclusions?

Reviewer #5: Partly

3. Has the statistical analysis been performed appropriately and rigorously? 

Reviewer #5: Yes

4. Have the authors made all data underlying the findings in their manuscript fully available?

Reviewer #5: Yes

5. Is the manuscript presented in an intelligible fashion and written in standard English?

Reviewer #5: Yes

6. Review Comments to the Author

Reviewer #5: (No Response)

7. PLOS authors have the option to publish the peer review history of their article (what does this mean?). If published, this will include your full peer review and any attached files.

Reviewer #5: No

---

## [Editor Report · Acceptance letter]

10 Jul 2024

PONE-D-23-29142R4 

PLOS ONE

Dear Dr. Franks, 

I'm pleased to inform you that your manuscript has been deemed suitable for publication in PLOS ONE. Congratulations! Your manuscript is now being handed over to our production team.

Kind regards, 

on behalf of

Dr. Daniel de Paiva Silva 

Academic Editor

PLOS ONE